

# A wave-resolving modeling study of rip current variability, rip hazard, and swimmer escape strategies on an embayed beach with irregular rip channels

Ye Yuan[1,2], Huaiwei Yang[1,2,*], Fujiang Yu[1,2], Yi Gao[1,2], Benxia Li[1,2], and Chuang Xing[1]

[1]National Marine Environmental Forecasting Center of China, Beijing, 100081, China
[2]Key Laboratory of Research on Marine Hazards Forecasting, Ministry of Natural Resources of China, Beijing, 100081, China

**Correspondence:** Huaiwei Yang (yanghw@nmefc.cn)

**Abstract.**

Drownings due to rip currents are a major threat to beach safety. In this study a high-resolution Boussinesq model with a modified wave-resolving Lagrangian tracking module has been applied to a 2-km-long embayed beach, Dadonghai of Sanya, Hainan Island, with the purpose to study rip current variability, real-time rip hazard identification, and the optimal swimmer escape strategies. Beach stage plays an important role in the occurrence and strength of rip currents. Satellite images of the Dadonghai Beach shows that crescentic bars and shore-connected transverse bars emerge alternatively in the embayed beach, which results in long-term modulation of rip strength periodically according to the modeling. A series of tests are designed and confirms that rip current strength is closely related to wave properties and tidal levels. Spectral analysis of output time series at specific points shows that rip currents fluctuate on the orders of 1 min and 10 min, which reflects the effects of wave-group and VLF motions. Real-time evaluation of rip hazard is crucial for providing the lifeguards and general public with appropriate information on the occurrence and location of the rip currents, and how to set patrolled area. However, fine-scale numerical modeling using the existing hydrodynamic models are computing-demanding due to a cascade of basin-scale meteorological and wave climate modeling down to the wave-resolving scale. In this study, an attempt of using GPU-accelerated Boussinesq model embedded with the spectral wave model (WAM6) has been made, which enables a faster and more complete description of rip hazard. Lagrangian tracking of virtual swimmers demonstrates that multiple factors contributing to the survival of swimmers caught in the rip currents, include surf-zone bathymetry, rip strength, flow patterns, and swimmer's position. For weak-to-moderate rip currents and longshore currents, *swim onshore* consistently seems the most successful strategy across all the scenarios in this study. For swimmers within the inner surf zone, the successful rate is satisfactory by taking strategies of either *swim onshore* or *swim parallel to the beach*. Higher surf-zone exit rate along the Dadonghai Beach are not favorable to *stay afloat* action, which put swimmers at a higher risk of being expelled to deeper water. Pulsation of rip currents in wide rip channels can form swirls or eddies which can also be hazardous to swimmers with weak swimming ability. One of the differences of the present study from the previous works is that the random, wave-resolving modeling was adopted for Lagrangian tracking of virtual swimmers with 1-m resolution. Virtual trajectories yielded by the wave-resolving and wave-averaged velocities are generally consistent with each other. However, using Boussinesq model shows its superiority in studying fine-scale





nearshore circulation and its variability, as well as understanding the effects of wave randomness and directional spreading on surf-zone flows.

## 1 Introduction

Rip currents are narrow jets of offshore-directed flow that originated in the surf zone. Strong rip currents can reach above speeds of 0.5 - 1.0 m/s and persist for minutes, thus taking swimmers of all ability levels into deeper water (Dalrymple et al.,

2011; Castelle et al., 2016b). It is reported that the majority of the bather drowning and beach rescue efforts worldwide are related to rip currents (Brighton et al., 2013; Arun Kumar and Prasad, 2014; Brewster et al., 2019; Castelle et al., 2020). In China, concerns on rip hazard was raised until recently as more coastal fatalities were reported (Li, 2016; Zhang et al., 2021). Taking Dadonghai Beach, Hainan Island as an example, more than one hundred rescues and 7 deaths have been recorded in less than a month from August 1 to 23, 2013, according to online report (Li and Zhu, 2018). Therefore, better understanding

on rip hazard is necessary for educating the public to avoid rip current-related drownings.

Wave breaking is the main driving force in surf-zone hydrodynamics by constantly producing onshore momentum and mass fluxes. Momentum and mass conservation is kept to balance these fluxes by introducing change in the mean water level near the shoreline, which provides a hydrostatic force, or a bottom friction force, due to wave-induced currents (Shepard, 1936; Bowen, 1969). It is widely accepted that rip currents are generated by alongshore variations on breaking wave heights. In the

bar-trough beach, an idealized rip current system is depicted as feeder currents, rip neck, exit flow, and rip head (MacMahan et al., 2006). Incident waves break over the sandbars, and result in the formation of longshore feeder currents usually flowing along or behind the sandbar. The feeder currents converge into a narrow and fast-flowing rip neck, and then exist through incised rip channels.

The nearshore circulation is temporally unstable and spatially variable, which makes the accurate identification and mod-

eling of rip current hazard a difficult task. Temporal variations induced by the changes in the incident wave conditions, tidal modulation, Very Low Frequency (VLF) motions, and infragravity motions, all contribute to the dynamic signature of the rip-current circulations (Reniers et al., 2010). Observations and modeling studies suggest increasing rip strength under higher wave forcing and lower tidal levels (Castelle et al., 2020). Oblique incident waves are inclined to generate strong alongshore currents, which deflects to offshore direction by headland or rigid boundary. Rip currents also pulsate at lower frequency as

response to incident wave groups. The shape of the morphology (i.e., orientation, width, and rip spacing) also determine the rip current patterns, scales and magnitude (Dalrymple et al., 2011). The modeling of the localised features require fine resolution of up to 1 meter, which used to be limited by computing power.

It is important to relate knowledge on rip current dynamics with rip hazard mitigation activities. To reduce rip current hazard and keep beach safety, a series of concerns need to be addressed: How and what level the rip hazard is in a specific beach, and

how swimmers caught in a rip should react to survive. There are multiple ways to quantify or forecast rip current hazard. For bathymetrically controlled rip currents, its hazard can be inferred based on the beach morphodynamic state model proposed by Masselink and Short (1993). However, as rip currents can be present during all the evolving beach stages, meteorological





and hydrodynamic factors should be included. By establishing empirical relationship among rip-related rescues, weather, wave condition, tidal level and other factors (Engle et al., 2002; Dusek and Seim, 2013), rip hazard levels can be indirectly predicted
for specific beach. Since the method is empirically based, derivation of a universal relation is questionable and its validity need to be calibrated individually. Recently, the framework of predicting rip hazard by numerical tools has been proposed and implemented in Korea (Kim et al., 2013; Eom et al., 2014). With the aid of High-Performance Computation (HPC) and GPU acceleration, high-resolution beach-scale modeling of wave-driven currents are practicable using meteorological and spectral wave forecasting as forcing conditions, and thus provide a promising way to assess real-time rip current hazard in a specific
region (Yuan et al., 2020).

    Another category of studies focuses on field and modeling studies of the optimum current escape strategies. Field studies require the participation of experienced swimmers capable of implementing different swimming strategies as instructed. Three typical swimming strategies tested in these studies are *stay afloat*, *swim parallel*, and *swim onshore*. Field tests organized by McCarroll et al. (2014) and Van Leeuwen et al. (2016) found that there were not versatile escaping strategy for swimmers
caught in the rip currents, as significant variations in environmental factors (i.e., beach morphology, wave conditions) and resulting rip current flow regimes exist. McCarroll et al. (2015) packed a swimmer tracking and safety evaluation module to the nearshore hydrodynamic model XBEACH based on depth-averaged shallow water equations. By seeding a number of swimmers in a single rip current system, a series of scenarios and sensitivity tests are implemented to seek a preferable escape strategy. The results showed that low-speed and continuous swimming may be more effective than floating, while the choice of
the best swimming direction is closely related to the starting position, rip spacing and more complex factors. Similar research has been conducted by Castelle et al. (2016a) on multiple rip channels along an open beach in France. Simulations showed that subtle changes in the bar-rip morphology had a large impact on the rip flow field, and in turn on the alongshore variability of the optimal rip current escape strategy.

    To date, most of rip hazard simulations have been performed by time-averaging the phase of gravity waves. Due to the
episodic and non-stationary nature of rip current flows, phase-resolving models, also known as Boussinesq-type model (BTM), are more preferable to study rip currents dynamics, and capture random, instantaneous trajectories of swimmers (Castelle et al., 2016a). In the study, a phase-resolving BTM accelerated by GPU, FUNWAVE-GPU (Shi et al., 2012; Yuan et al., 2020), is used to explore rip current variability, rip hazard and swimmer escaping strategies in Dadonghai, an embayed sandy beach with irregular rip channels at Sanya, Hainan Island. A four-grade fine-scale rip hazard map based on the numerical assessment
of rip-current strength and duration is presented. The swimmer tracking and safety evaluation module proposed by McCarroll et al. (2015) is coupled to FUNWAVE-GPU to explore the swimming escape strategies. Discussion has been made to highlight how the phase-resolving and random trajectories of swimmers are different with those using time-averaged velocities. The study is a part of ongoing efforts to construct rip hazard forecasting and mitigation system for rip-prone beaches in China.

    The paper is organized as follows. Section 2 provides an introduction on study site in terms of hydrodynamic background
and beach morphology. In Section 3, numerical approaches on rip hazard grading and swimmer escape strategies are briefly described. The results are presented in Section 4, with further discussion on phase-resolving and time-averaged tracking of swimmers given in Section 5. Conclusions are made in Section 6.



## 2   Study site

The study site is Dadonghai beach, which is a medium-energy, micro-tidal, embayed beach in the south tip of Sanya, Hainan
Island (1). The beach faces south with a horizontal span of roughly 2 km. Tourists flock to Dadonghai due to its preferable
climate and clear sand throughout the year, and most of them are from inland China and foreign countries. The daily tourist
reception reaches more than 5,000 per day during the peak period. The study site was selected as it is marked as a high-risk
beach with rip drownings reported annually.

### 2.1   Surf-zone bathymetry

Two satellite images were acquired from Google Earth Historical Database on August 7, 2018 and December 26, 2019 when sea
state was calm. The images feature a typically complex beach planform composed of crescentic/transverse bars and rip channels
with irregular configurations. A mild slope with straight and parallel contours exists seaward of the surf zone. As highlighted
in Fig. 1b-e, two sets of the surf-zone bathymetry suggest slightly different beach stages. The former exhibits shore-connected
transverse sandbar with incised rip channels. White foam corresponding to breaking waves highlights the presence of shallow
bars, and darker areas represent deeper rip channels that penetrate through the bars. While in the subsequent image collected
in 2019, the rhythmic crescentic bar with wider yet shallower rips can be observed. Compared with the former morphology, it
is supposed to generate weaker rip currents (Wright and Short, 1984).

Hourly mean tidal elevation was obtained from the nearby tidal gauge. Generally, Dadonghai exhibits a mixed semi-diurnal
tidal cycle, with a micro-tidal range varying between 1.0 - 2.0 m throughout the year. The largest tidal range occurs from
October to January. The west end of the bay is shallow and has coastal coral reef beneath the surface, while the east end is
characterized by bedrock covered with a mixture of cobbles and boulders, suggesting high wave-energy environment. During
the low tide, both ends is exposed, which can also been observed in the satellite images (Fig. 1b, d).

Compared to extensive sonar or in-situ measurements of depth, shallow bathymetry can be fast and cost-effectively evalu-
ated by remote-sensing images. By establishing a site-specific linear relationship between pixel colors and depths, nearshore
bathymetry at Dadonghai was mapped to 1-meter resolution. The derived bathymetry was rotated 90 degrees to align the
shoreline with the vertical axis. The rotation is necessary for FUNWAVE to apply irregular wave maker and periodic boundary
condition. Although this inversion may not produce bathymetry as accurate as other approaches, it can be operationalized for
rip hazard forecast in future owing to its simplicity to locate sandbars and shoals, as well as availability of satellite imagery
(Radermacher et al., 2018).

### 2.2   Wave conditions

Based on an analysis of 30-year wave hindcast dataset developed by National Marine Environmental Forecasting Center of
China (NMEFC), waves during the summer months are relatively more energetic than in winter off Sanya city, which is
interspersed with high-energy events associated with tropical cyclone activities in the northern South China Sea (SCS). The
1-day moving averaging on the hourly wave hindcast in 2018 is performed and shown in Fig. 2a. The average significant wave



height during the summer season was around 0.7 m with a mean wave period of 5 - 8 s. According to wave rose diagrams in Fig. 2b-c, the prevailing wave directions in Dadonghai Beach were from south and southeast. Due to its orientation, it received stronger swell in summer due to the prevailing Asian monsoon winds from southwest. Two prominent peaks during Julian days of 195 - 205 and 255 - 265 were explained by typhoons moving westward through the Northern SCS, leading to elevated significant wave height of 1.5 - 2.5 and 2.0 - 4.0 m (without 1-day moving average), respectively. Strong swell could persist for

a week. Under normal weather conditions, the daily-averaging significant wave height could reach up to 1.0 m sporadically, which was attributed to swells propagating into the bay from open sea.

## 3  Methods

### 3.1  Numerical model

Phase-resolving Boussinesq-type wave models have proven to be robust tools for modeling surface waves and wave-driven

processes in the nearshore region (Shi et al., 2012; Chen et al., 1999, 2003; Geiman et al., 2011). In this paper, we use the FUNWAVE-TVD (total variation diminishing version of the fully nonlinear Boussinesq wave model) to simulate rip current dynamics. Due to the existence of high-order dispersive terms, FUNWAVE-TVD is more computationally demanding compared with shallow water equation solvers (Kirby, 2016). To address this problem, a multi-GPU-accelerated version of FUNWAVE-TVD (FUNWAVE-GPU) has been developed recently.

In our simulation, random, directionally incident waves are generated, propagate shoreward, and then shoal, break, finally produce wave setup in the surf zone. The computation domain is $1678 \times 659$ m$^2$. The cross-shore and alongshore grid size is chosen to be 1 m with variable time step determined by CFL stability condition. The fine resolution is required to resolve wave-induced flow behavior with different scales. In the study time step is usually smaller than 0.04 s, which is necessary to resolve individual wave. The bottom friction coefficient is 0.0025. The directional irregular wavemaker is placed at 10-m water depth

offshore, which is approximately 500 m away from the shoreline. The wavemaker can generate random, normally or obliquely incident waves with specified peak amplitude, period and incident angle. The absorbing boundary conditions are placed behind the wavemaker. The offshore sponge layer has a width of 100 m and is used to absorb outward-propagating waves. Wave breaking is modeled by two schemes implemented in the FUNWAVE; either the shock-capturing scheme of Tonelli and Petti (2009), or the eddy-viscosity scheme following Kennedy et al. (2000). The latter scheme is used in the study. The wave-driven

flow field is obtained by averaging the instantaneous fluid particle velocity over two wave periods.

Besides, unlike previous studies on rip current dynamics and bather tracking using wave-averaging models with coarser grid resolution, this study provides a chance to showcase to what extent the tracking can be different with wave-resolving and wave-averaged flow velocities, respectively.





**Table 1.** Model input for sensitivity tests (T1-T9) of rip currents on incident wave conditions and tidal levels.

| Tests | $H_{peak}$ (m) | $T_{peak}$ (s) | $\lambda$ (°) | Tide (m) |
|-------|----------------|----------------|---------------|----------|
| T1 | 0.7 | 12.0 | 0 | 0 |
| T2 | 1.0 | 12.0 | 0 | 0 |
| T3 | 1.3 | 12.0 | 0 | 0 |
| T4 | 1.8 | 12.0 | 0 | 0 |
| T5 | 1.0 | 4.5 | 0 | 0 |
| T6 | 1.0 | 8.0 | 0 | 0 |
| T7 | 1.0 | 12.0 | 5 | 0 |
| T8 | 1.0 | 12.0 | 20 | 0 |
| T9 | 1.0 | 12.0 | 0 | +0.6 |

## 3.2 Hydrodynamic settings

Prediction of nearshore circulation are important for swimmer safety and for estimating surf-zone dispersion of sediments and pollutants. Rip currents are forced by incoming waves, and modulated by tidal elevation and other low-frequency motions MacMahan et al. (2006). In the study, hydrodynamic response of rip currents is examined by a series of numerical tests with varying offshore wave forcing conditions and tidal elevations. Shallow bathymetry within the surf zone largely dictates where incident waves break, thus two sets of bathymetry shown in Fig. 1b-e are also applied. In each simulation, the model runs for 160 50 min, with the first 10 min neglected due to a cold startup.

Model input is summarized in Tab. 1. The peak significant wave height and period range from 0.7 m to 1.8 m and from 4.5 s to 12.0 s, respectively. The incident angle varies from shore-normal direction (0°) to obliquely incident (20°). Tidal elevation is considered by adjusting input bathymetry slightly. These tests are representative of summer wave conditions, and hereafter referred to as T1 - T9.

## 3.3 Rip Hazard Levels

We define rip currents as the offshore-directed flow with direction values falling between 135 and 225° clockwise from North. The rip strength is divided into 4 intervals shown in header row of Tab. 2. For each rip strength interval, its duration is the accumulated time period ($t_{rip}$) when the velocity falls into it during the entire simulation period ($t_{modeling}$). Rip duration is simply measured by $t_{rip}/t_{modeling}$, and categorized as 4 levels listed in the first column of Tab. 2. To quantify rip hazard, 170 here we propose the term of Rip Hazard Level, and define it as a combination of rip strength and duration. The classification is summarized in Tab. 2. Grade I denotes the highest Rip Hazard Level potentially posing greatest danger to the bathers, and Grade IV is the lowest. Rip Hazard Levels are evaluated at location where the water depth is deeper than 0.8 m. Below this





**Table 2.** Classification of rip hazard based on the rip strength (m/s) and rip duration ($\frac{t_{rip}}{t_{modeling}}$).

| Rip duration ($\frac{t_{rip}}{t_{modeling}}$) | Rip strength (m/s) | | | |
| --- | --- | --- | --- | --- |
| | $V_{max} < 0.3$ | $0.3 \leq V_{max} < 0.6$ | $0.6 \leq V_{max} < 0.9$ | $V_{max} \geq 0.9$ |
| $0.05 \leq t_{rip}/t_{modeling} < 0.1$ | 1 | 2 | 3 | 4 |
| $0.1 \leq t_{rip}/t_{modeling} < 0.2$ | 2 | 4 | 6 | 9 |
| $0.2 \leq t_{rip}/t_{modeling} < 0.4$ | 3 | 6 | 9 | 12 |
| $t_{rip}/t_{modeling} \geq 0.4$ | 4 | 8 | 12 | 16 |

**Rip Hazard Level: Grade IV $\in$ [1,4]; Grade III $\in$ [5,8]; Grade II $\in$ [9,12]; Grade I $\in$ [13,16]**

depth, it is assumed that an adult of average height is capable of standing firmly in the water even with high-energy wave condition.

### 3.4 Phase-resolving tracking of swimmers and safety criterion

The existing Lagrangian tracking module of FUNWAVE has been modified to simulate movement of swimmers with a combination of instantaneous, random wave motion ($\tilde{u}_w$) and swimming velocity ($U_s$), as shown in Eq. 1. The effect of individual wave on swimmers can be resolved due to the phase-resolving nature of Boussinesq type wave model. Swimmers are initially seeded as particles in the surf zone with the wave orbital motion and the mean depth interpolated from the neighboring four 180 grid points at each time step $\triangle t$. Each particle is also assigned with a fixed swimming velocity and direction at each $\triangle t$. An arbitrary factor of 0.8 accounts for the correction of drifting speed.

$$u_{tracking} = 0.8\tilde{u}_w + U_s. \tag{1}$$

The safety check of swimmers follows the work of McCarroll et al. (2015), which established a local hazard rating criterion $HR$ (Eq. 2) to check whether swimmers have reached safe state.

$$HR = \overline{d}(\overline{U} + 0.5), \tag{2}$$

where $\overline{d}$ is the mean water depth, and $\overline{U}$ is the wave-averaged flow velocity. As shown in Table 3, a successful escape should satisfy that the swimmer is at a position where either the mean water depth or the $HR$-value is below a threshold ($d_{safe}$, or $HR_{safe}$). The rip escape simulation is performed in the domains indicated in Fig. 1 by red rectangles. The swimmers are seeded in the rip channels uniformly with a spacing of 5 m when the modeling reaches a steady state. Although elite swimmers 190 can propel themselves at up to 1 m/s in still water, here we assume that the average swimmers caught in rip currents have a swimming velocity of 0.2 - 0.4 m/s. Swimmers are supposed to be exhausted, and thus removed from the subsequent simulation when a maximum time period of 10 minutes is exceeded. The escape time ($t_{safe}$) is recorded after each swimmer reaches the safe state. Three escape strategies, as suggested by McCarroll et al. (2015) and Castelle et al. (2016), are tested in the study, including *stay afloat*, *swim onshore*, and *swim parallel to shoreline*. For *stay afloat* strategy, $U_{swim}$ equals to zero, suggesting





**Table 3.** Configurations of rip escape strategies and safety check criterion

| Swimming Strategies | | Adult | | Child | |
|---|---|---|---|---|---|
| | | $U_{swim}(m/s)$ | Safety check | $U_{swim}(m/s)$ | Safety check |
| Stay afloat | | 0 | | 0 | |
| Swim onshore | | 0.2, 0.4 | $HR_{safe} = 0.7m^2/s$ | 0.2 | $HR_{safe} = 0.5m^2/s$ |
| Parallel to shoreline | westward | 0.2, 0.4 | $d_{safe} = 1.1m$ | 0.2 | $d_{safe} = 0.7m$ |
| | eastward | 0.2, 0.4 | | 0.2 | |

swimmer completely move with ambient wave motions. Table 3 provides a summary on rip escape strategies and safety check criterion. Adult and child have different swimming capability and safe check conditions.

# 4 Results

## 4.1 Hydrodynamic response to wave conditions and surf-zone bathymetry

By applying two sets of beach morphology obtained in 2018 and 2019 in the simulations, the snapshot of nearshore circulation
forced by different wave climates and tidal conditions are shown in Fig. 3-4. In Fig. 3a-d, the most prominent feature is that two opposite embayment-scale longshore currents originate from both ends of the bay with maximum flow velocity reaching 0.9 m/s. The incident waves shoal and break at both headlands immediately after entering the embayment, and in turn produce alongshore variable wave setup creating longshore currents. The presence of strong lateral shear in the cross-shore direction leads to the meandering of the longshore flow, which gradually deflect to offshore direction at $x = 600$ m and 1400 m. Swimmers
who caught in these mega rips may exit the surf zone rapidly. Rip current dynamic within the red rectangle is enlarged in Fig. 3e-l. The incised rip channels with uneven spacing produce complex surf-zone flow regime. Generally, rip currents get stronger with increasing incident wave height. For test case $H_s = 1.0$ m and $T_p = 12$ s, a intense shore-normal rip current ( 0.6 m/s) at $x$ = 900 m is generated with stable feeder currents from neighboring sandbars. This well-established rip extends more than 150 m offshore, and persists over the entire simulation. The animation shows considerable vibration of rip direction and accompanied
vortex structures.

Nearshore circulation is weaker in terms of extent and strength for incident waves with shorter period. Dadonghai Beach faces the Northern SCS. While the period of incoming waves are generally within 4-8 s, the long-period swells with period of 8-14 s occasionally propagate into the embayment during the typhoon season. Averagely, there are more than 13 tropical cyclones (TCs) across the Northern SCS annually from early March to later November. Swells arrive days before the landing/passing of
TC and usually last for more than a week. Even subtle variation in incident direction can induce considerable transition of flow pattern. Offshore-directed flows are suppressed and rapidly deflected to longshore direction.

The surf zone of sandy beaches usually shows a variety of complicated morphological patterns appearing alternatively with time. The beach morphology on December 26, 2019 exhibits a different beach stage. The satellite image (Fig. 1d-e)





shows a nearly straight shore-parallel sandbar approximately 80-100 m seaward from the shoreline with periodic horn-shape

bars weld to the shore, causing the discontinuity of alongshore trough between the shoreline and outer sandbar. This state is

known as crescentic bar. Compared with previous beach state in 2018 (Fig. 1c-d), there is no typical rip channels which are

characterized by penetrating exit openings. The presence of relatively small bathymetric variations can have a profound effect

on rip circulation. In this case, weaker exit flow is expected. As shown in Fig. 4a-h, the circulation is mainly confined within

the surf zone, and no persistent and well-established exit flow is formed. The animation also suggests rip flow patterns consist

of semi-enclosed vortices developed within the wide channels, resulting in spatial and temporal variability in flow strength and

direction.

We use rip-current rose diagram to interpret the temporal distribution of rip current speed and direction over the entire

simulation period. The time series of rip current at Gauge A ($x$ = 730 m, $y$ = 120 m, Fig. 1f) is interpreted by rose diagram in

Fig. 5. The length of each colored spoke is a measure of the percentage of time that the rip current flowing to that particular

direction. For moderate wave energy with $H_s$ = 1.0 m, the rip is nearly shore-normal with magnitude varying between 0.3 - 0.6

m/s. The rip strength can exceed 0.6 m/s occasionally with $H_s$ = 1.3 m. The wave-driven flow is weaker and shows considerable

variability in direction when incoming waves have shorter period. Tidal modulation of rip current has long been confirmed by

a number of observational and modeling studies (Dalrymple et al., 2011; Castelle et al., 2020). Generally rip strength is well

correlated to tidal level, with maximum rip currents occurring at low tide. In Fig. 5f, by deepening the bathymetry with a

constant of 0.6 m, rip current is replaced by a weak, and meandering alongshore flow regime that is coupled to the underlying

surf-zone morphology.

Temporal variation of rip currents is closely related to the forcing mechanisms and local bathymetry features (Reniers et al.,

2010). We placed a group of gauges at well-formed sandbars and rip channels along the shore with output interval of 2 s. The

majority of gauges show periodic fluctuations on flow magnitude and direction at multiple temporal scales. Fig. 6 illustrates

analysis on time series of Gauge B within a rip channel ($x$ = 1060 m, $y$ = 105 m of bathymetry on August 7, 2018). The rip

flow exhibits intermittent, periodic nature with flow direction oscillating from side to side in the channel. Normalized power

spectra of the modeled current (gray line) are plotted in subplot (b). The spectra shows a broad-banded feature, with two

prominent peaks occurring at $T$ = 80 - 100 s and 8 - 10 min, which correspond to period bands of infragravity pulsations

and Very Low Frequency (VLF) motions, respectively. The incident wave groups have been known as the main cause of rip

fluctuations at infragravity band (MacMahan et al., 2004a, b). The time series of incident wave height confirms the existence of

wave group effect, and its power spectrum (dashed line in Fig. 6b) shows an energy peak at 80 - 100 s. Within an wave group,

arrival of higher waves break and produce greater wave setup over bars, which results in stronger pressure gradient, and expels

excess water offshore through an intensified rip flow. This effect contributes to velocity variation of 0.1 - 0.2 m/s in Gauge

B. The presence of VLF motions can not be explained by wave forcing due to the absence of spectral peak at corresponding

period band. By observing rose diagram and vector plot of rip flow in subplot (c-d), the VLF motions are characterized by the

periodic shift of flow direction, which is usually related to the formation of vortex due to local morphology. Besides, while

infragravity wave-group fluctuation is permanent for all gauges, VLF motions are only present in some of them. It is expected



that swimmers in a weak rip current system at a specific time may be caught in strong flow soon afterwards due to the sporadic feature of rip currents.

## 4.2   Rip hazard maps

Using the rip hazard index table proposed in Section 3.3, rip hazard is interpreted quantitatively by an integration of rip strength and duration at each grid point. Therefore, Rip Hazard Levels are representative of both offshore-directed flow velocity and its persistence. In Fig. 7, spatial distribution of Rip Hazard Level is overlapped with the satellite images acquired on August 7, 2018 and December 26, 2019. Grid points with depth less than 0.8 m are masked.

As shown in Fig. 7, strong rip currents originate from the rip channels or troughs overall. A close examination of left panel finds that the area of white foam is staggered with rip necks. Rip hazard with transverse-bar morphology (Left panel) can produce higher hazard level of Grade-II. Strong rips can potentially eject swimmers seaward rapidly with the maximum extent of 100 - 200 m. Due to the existence of mega-rip originated from the deflected longshore currents at both flanks of the embayment ($x$ = 400 and 1400 m in Fig. 3-4), the area of Grade-III rip hazard spreads more than 200 m seaward. Swimmers who are caught into the deflected longshore currents may exit the surf zone unawares.

In the study, by using the Nvidia A100 graphic card, a maximum of speedup of 8-10 fold is achievable using FUNWAVE-GPU, compared with a 36-core Intel CPU node. A modeling of beach-scale wave propagation, breaking and associated nearshore circulation can be completed within 10 min (Yuan et al., 2020). Recently the WAM6 spectral wave model (Group, 1988) has been modified and accelerated by OpenACC by the authors (WAM6-GPU, source code can be accessed upon request). The computation time for a 5-day 1/12° wave modeling of the Chinese offshore regions (0 - 45° N, and 95 - 135° E) on Nvidia A100 card is dramatically reduced to approximately 5 min. By feeding the FUNWAVE-GPU with 2D directional spectrum data obtained from the basin-scale spectral wave modeling, or simply specifying peak wave parameters and tidal levels, it is possible to generate real-time rip hazard maps with fine resolution when nearshore bathymetry is available. It is helpful for beach safety practitioners to deploy rescues appropriately.

## 4.3   Swimmer escape simulations

Virtual swimmers seeded in Area 1 and Area 2 with different escape strategies are traced by instantaneous, wave-resolving velocity with wave forcing condition of $H_s$ = 1.0 m and $T_p$ = 12 s. The modeling results are shown in Fig. 9-12. Area 1 is a wide rip channel located between 860 m and 1000 m of the longshore axis, as marked in Fig. 1f. The water depth is slightly shallower at the seaward exit of the channel. According to Fig. 4, Area 1 does not contain a typical underlying morphology to incubate strong rip currents. The flow magnitude and direction show a large degree of variability due to the formation of a strongly asymmetric Counter-Clockwise Eddy (CCE) within the channel during the simulation, which is supposed to retain swimmers within the surf zone for a longer time. Area 2 is characterized by several parallel longshore isobaths without obvious rip cell. Stable longshore currents exist in this area, which gradually deflect to offshore direction at $x$ = 500 - 600 m. Due to the fact that rip currents are unstable and oscillate in multiple time scales, a series of tracking simulations are conducted with virtual swimmers seeded at different model times with an interval of 150 s. As suggested by Fig. 8 , the modeled trajectories



of the virtual swimmers show considerable variability when choosing different seeding times, which results in variations of escape time $t_{safe}$ histograms accordingly. $t_{safe}$ hereafter is an average of all simulations with different seeding times. Besides, we also selected a typical rip channel for modeling study of swimmer escape. As the conclusion was similar with the previous studies (McCarroll et al., 2015; Castelle et al., 2016a), the modeling result was not included in this manuscript.

### 4.3.1 Stay afloat

Swimmers adopting *stay afloat* strategy are subject to ambient current field (McCarroll et al., 2015; Castelle et al., 2016a). In Area 1, rip flow is relatively weak in strength, and oscillates in direction due to the existence of a recirculating flow regime. The majority of floaters are trapped in the eddy, with their trajectories exhibiting complex vortical patterns. Most of them eventually exit the surf zone after 10-min floating. Only a few of floaters are transported to the western sandbar within 10 min, where it is safe to stand and walk to shore for adults (Fig 9a-b). In Area 2 where the intense longshore flow is dominant due to the presence of headland with submerged coral reefs, over 90% of the floaters are swept eastward along the isobaths and eventually expelled offshore at $x$ = 400 - 500 m (Fig. 9e-f). The swimmers who are not aware of this strong longshore jets may be gradually dragged away from the surf zone unconsciously. Generally, adult has slightly higher chance to survive than Child.

Swimming and floating actions are two sides of the same coin when swimmers notice of being caught in a rip. Swimming against the flow may result in muscular fatigue and cramp, especially for beginners; while adopting floating strategy can save energy and increase the possibility of being rescued. For rhythmic bar-trough beaches, floating or swimming parallel are supposed to be reasonable choices, which help the swimmers escape from the rip jets and reach the proximate sandbars safely. However, according to the modeling study of Area 1 and 2, using the strategy of *stay afloat* alone is obviously an unwise action, suggested by over 90% of failure rate. The alongshore variability of underwater topography increases the complexity of flow regime. There is no versatile escape strategy even in the same beach.

### 4.3.2 Swim onshore

Swimming onshore is thought to be an instinct to survive from the rip currents for most of swimmers. In Area 1, 100% of skilled swimmers with onshore swimming velocity $U_s$ = 0.4 m/s can reach safety in less than 8 min (Fig. 10b, e). The $t_{safe}$-value depends on the distance of swimmer to safety depth ($d_{safe}$ = 1.1 m for adult), as well as their initial position within the surf zone. It takes longer time for swimmers seeded between $x$ = 920 and 940 m to reach the safety depth as they need swim against the CCE inside the trough. Any hesitation or interrupted swimming may lower the chance of survival even for skilled swimmers. The recirculating flow is hazardous to Child due to their weaker swimming ability and longer distance to safety depth ($d_{safe}$ = 0.7 m). On average, the failure rate is over 50%. Most of failures occurs at rip neck and downdrift of rip eddy. In Area 2, almost all the bathers with different swimming abilities can reach safety within a relatively short time. Obviously for bathers caught in a longshore current, swimming onshore is the best choice. Generally, adopting strategy of *swim onshore* seems quite successful. However, for swimmers caught in the offshore-directed jet of a typical rip-flow system, a combination of actions of swimming parallel and onshore should be a more optimal strategy than swimming onshore alone.





### 4.3.3 Swim parallel to shore

The successful rate by taking *swim parallel to shore* actions is largely subject to the specific locations of bathers and swim
direction relative to the current. As shown in Fig. 11, for bathers seeded in CCE-dominated Area 1, only those within the inner
surf zone can reach the downdrift sandbar and are able to stand firmly by themselves. However, some of beginning bathers or
children who swim westward across the CCE are swept offshore a bit by the meandering rip current, and then fail to reach the
western sandbar (Fig. 11a, c). When caught in the rip currents, the first priority is to escape from the jet as soon as possible.
In Area 1, some beginning bathers who swim eastward against the flow are at greater risk of being stuck in the CCE, which is
clearly observed by the circular trajectory in Fig. 12a. Though these bathers are seeded in the inner surf zone, they fail to reach
safety within 10 min by choosing an inappropriate swimming direction. In Area 2 (Fig. 11g-i), the eastward longshore current
is gradually intensified and deflected offshore from $x$=200 m to 400 m. Over 60% bathers with above-average swimming ability
can reach the safe depth by swimming against the flow. All the bathers that swim eastward are quickly carried away by the
deflected current, and the result is not shown in Fig. 12. Taking the strategy of *swim parallel to shore* alone can not increase
the rate of survival in the study. The odds are even heavily against the bathers who choose a wrong direction.

## 5 Discussions

### 5.1 Escape strategy summary

The nearshore circulation along the 2 km-long Dadonghai Beach consists of steady longshore currents originated from both
headlands, and multiple rip cells pulsating in strength at different time scales. Both types of flows can be hazardous. For
swimmers within the inner surf zone, either *swim onshore* or *swim parallel to the beach* can be a wise strategy to escape
from the rip flow, while *stay afloat* action may carry the swimmer further away the inner surf zone. Generally a typical rip
current ranges from 10-30 m wide. Even a beginning swimmer can reach the neighboring sandbars within 2 min (Figs. 11-12).
However, for swimmers located in the outer surf zone, the chance of survival decreases substantially by taking *swim parallel*
actions alone. Although many rip hazard outreach activities advocate *swim parallel to the beach* as the primary escape strategy,
*swim onshore* seems the most successful strategy across all the scenarios in this study. An average swimmer is capable to reach
the safe depth even from the outer surf zone by sustained strokes (Figs. 10). It should be note that the rip strength is moderate
(0.2 - 0.4 m/s on average) for the specified wave conditions in AREA 1. Otherwise, the modeling results of Castelle et al.
(2016a) indicate that failures of the *swim onshore* occur in the rip neck where swimmers are stuck in the channel due to strong
offshore-directed jet.
*Stay afloat* action can be a viable and energy-saving strategy if the swimmers are within the surf zone with lower exit rate.
In this case, most of the swimmers can drift with the circulation cells and remain nearshore. MacMahan et al. (2010) reported
that only 19% of the wave drifters deployed in the rip currents exited the surf zone per hour. A 2-day observations conducted
by Gallop et al. (2018) found that the exit rates, however, is highly variable from 6% to 71%, depending on the incoming wave
breaking and pulsation of surf-zone currents. The results of this study indicate that the surf-zone exit rate is significant higher



in this embayed beach, and adopting *stay afloat* action alone is not a wise strategy, with a high risk of being expelled to deeper water (Castelle and Coco, 2013).

Besides, pulsation of rip currents in wide rip channels can form swirls or eddies which can also be hazardous to beginning swimmers or children, especially for those swimming eastwards against the CCE in AREA 1. Failure rates of 60% and 78% are observed for beginning swimmers ($U_s = 0.2$ m/s) and children. This suggests that space between sandbars are also an
important factor to be considered. In longshore current-dominated nearshore area, *swim onshore* strategy is undoubtedly the optimal action. The mean duration to safety is only 1.2 min for an average swimmer.

## 5.2 Lagrangian tracking by wave-resolving and wave-averaged velocities

In FUNWAVE-TVD, the random directional wave field is generated by using an internal wavemaker. It is basically an interior source term which integrates wave components split by frequency, direction and with random phases (Wei et al., 1999). The
input for the wavemaker can be either wave bulk parameters (i.e., peak wave height and period), or TMA shallow-water spectrum. In the study, swimmers are tracked by instantaneous velocity at the time interval $\triangle t$ of 0.04 s, rather than the wave-averaged velocity (i.e., averaging phase-resolving velocity over 24 s). It results in jagged trajectories shown in Fig. 9-12. Due to the random nature of wave field, the trajectory of each seeded swimmer is composed of a slow meander that exhibits mean flow pattern, and a much faster random oscillation at a wave time scale. One remaining question is whether wave-resolving
and wave-averaged tracing of individual swimmer can reproduce consistent trajectories, and whether the difference between the trajectories can be ignored so that it do not affect the conclusion of escaping modeling in the study. In Fig. 13, a 10-min tracking of virtual swimmers by instantaneous and wave-averaged velocity is illustrated with two overlapped curves (light and dark gray lines). The ends of each pair of trajectories are connected by orange lines. Fig. 13a-b suggest that resolving wave scale in the Lagrangian tracking is not necessary, and using Lagrangian mean velocity yields comparable results of escape time
($t_{safe}$) from using the wave-resolving velocity (Fig. 13c-d).

In the addition, previous modeling studies in rip-current escape strategies (McCarroll et al., 2015; Castelle et al., 2016a) mainly adopt short-wave-averaged model, in which the current field is assumed to vary more slowly than the short-wave period, and the rotational component of wave forcing on the current is given as a divergence of the radiation stress tensor (Longuet-Higgins and Stewart, 1962). While in this study the wave-resolving FUNWAVE is used. Each type of model has
been shown to reproduce similar spatial distribution of time-averaged wave height and coherent vortex structures (Geiman et al., 2011; Terrile et al., 2006) in the surf zone, thus can be used to carry out rip-current-related studies.

## 6 Conclusions

This paper presents the results of a Boussinesq modeling of rip hazard and escape strategies in an embayed recreational beach with grid resolution of 1 m. Especially, discussions have been made on the variability of rip currents and how wave-resolving
tracking of virtual swimmers differs from that using wave-averaged velocity.



Beach stage plays an important role in the occurrence and strength of rip currents (Dalrymple et al., 2011). Surf-zone bathymetry obtained in 2018 and 2019 exhibits different beach stages. The modeling results show that shore-connected transverse bars with incised, narrow rip channels are favorable to strong rip cells, while the crescentic bars separated by rip troughs with shallow exits, though, generate more complex yet slightly weaker nearshore circulations. A series of Boussinesq modelings indicate that rip current strength is closely related to several incoming wave properties, including wave height, peak period, and incident angles. Tidal level also exhibits the modulating effect on rip strength. The results agree with the previous studies on variability of rip currents. Spectral analysis of the output time series shows that rip currents fluctuate on the orders of 1 min and 10 min, which reflects the effects of wave-group and VLF motions. The pulsation of rip currents has importance for beach safety.

Quantitative estimate rip hazard has been enabled by the high-resolution modeling of nearshore circulation. In this study we defined a four-level rip hazard indexes based on a combination of rip flow strength and its duration. With the state-of-art GPU computation facility, operational forecasting of rip hazard level now is possible within 20 min based on the FUNWAVE-GPU embedded to GPU-accelerated spectral wave model.

Outcomes of the modeling demonstrate that multiple factors contributing to the survival of swimmers being caught in the rip currents, include surf-zone bathymetry, rip strength, flow patterns, and swimmer's position. Considering the temporal and spatial variability of nearshore circulation, neither strategy is 100% successful, and a combination of different actions is necessary for specific occasions. For weak-to-moderate rip currents and longshore currents, *swim onshore* consistently seems the most successful strategy across all the scenarios in this study. For swimmers within the inner surf zone, the successful rate is satisfactory by taking strategies of either *swim onshore* or *swim parallel to the beach*. To reduce risk at the shoreline, simply educating the public to stay inside the inner surf zone and enter water with buoyancy is a basic rule. *Stay afloat* action can be a viable and energy-saving strategy if the swimmers are within the surf zone with lower exit rate. However, Floating generally resulted in longer times to safety with higher variability compared to swimming parallel or onshore. The surf-zone exit rates are high variable depending on individual beach stages and incoming wave conditions. Our results indicate that adopting *stay afloat* action alone may lower the chance of escape, and suffer a high risk of being expelled to deeper water. Besides, pulsation of rip currents in wide rip channels can form swirls or eddies which can also be hazardous to swimmers with weak swimming ability. This suggests that space between sandbars are also an important factor to be considered. Psychological factors also contribute to the rip escape, which is not discussed in the study.

There are two primary types of models that are used to simulate surf zone flows. While the majority of the rip-current studies used short-wave-averaged model, in this study we used the Boussinesq model which required higher grid resolution to resolve wave crests. Inter-comparison between wave-resolving and wave-averaged tracking of virtual swimmers demonstrates that both approaches yield comparable modeling results. However, using Boussinesq model is helpful in studying fine-scale nearshore circulation and its variability, as well as understanding the effects of wave randomness and directional spreading on surf-zone flows.



*Author contributions.* The manuscript was written by Ye Yuan. FUNWAVE-TVD was GPU-accelerated by Ye Yuan when visiting University of Delaware in 2019. The computation and visualization were carried out by Huaiwei Yang. Fujiang Yu conceived the research. Surf-zone bathymetry was mapped by Yi Gao using Satellite images obtained from Google Earth. Benxia Li and Chuang Xing processed the wave-buoy data.

*Competing interests.* The authors declare that they have no conflict of interest.

*Acknowledgements.* The work is supported by the Innovative Youth Talents Program, MNR of China, as well as National Science and Technology Major Project of China (Grant no. 2016YFC14015).



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





**Figure 1.** Location map of Dadonghai Beach, Sanya (a), with satellite images collected on August 7, 2018 and December 26, 2019 (b-e). The satellite-derived bathymetry contours with spatial resolution of 1 m are displayed with orientation rotated 90 degree clockwise (f). Gauge A and Gauge B are set to investigate rip current variability. Hypothetical swimmers are evenly seeded within Area 1 and 2 marked by two red rectangles, with the purpose of studying swimmer escape strategies. The image source is Google Earth.

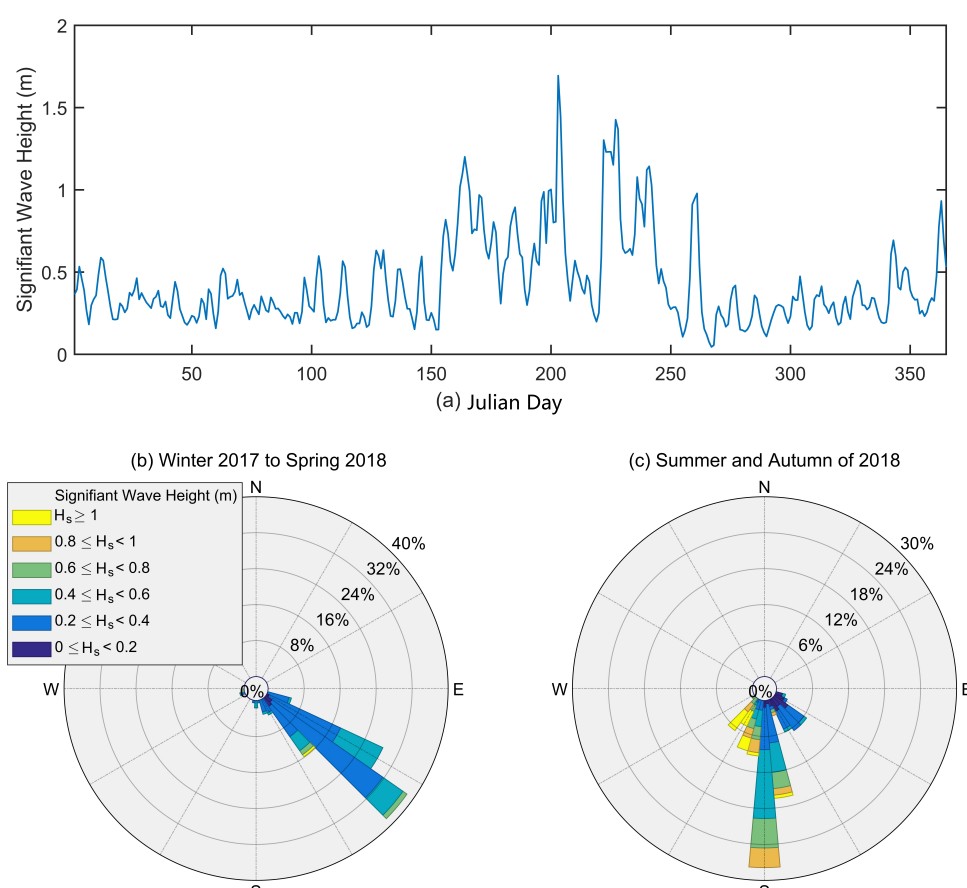

**Figure 2.** Wave climate in study site. Subplot (a) in the above panel is 1 day-averaged time series of significant wave height in 2018 by NMEFC's wave hindcast dataset; (b-c) are wave rose diagrams for two different monsoon seasons.




**Figure 3.** Snapshot of wave-driven velocity field in the embayed Dadonghai Beach with different forcing wave conditions labeled in the top of each subplot. Rip flows within the red rectangle in subplot (a-d) is enlarged in (e-l). Bathymetry is acquired on August 7, 2018 (corresponding to Fig. 1b-c, and characterized by transverse bars incised by relatively deep rip channels. Depth contours are overlapped as thick, black lines.





**Figure 4.** Snapshot of wave-driven velocity field in the embayed Dadonghai Beach with different forcing wave conditions labeled in the top of each subplot. Bathymetry is acquired on December 26, 2019 (corresponding to Figure 1d-e), and characterized by crescentic outer bars with periodic shallow and wide troughs.

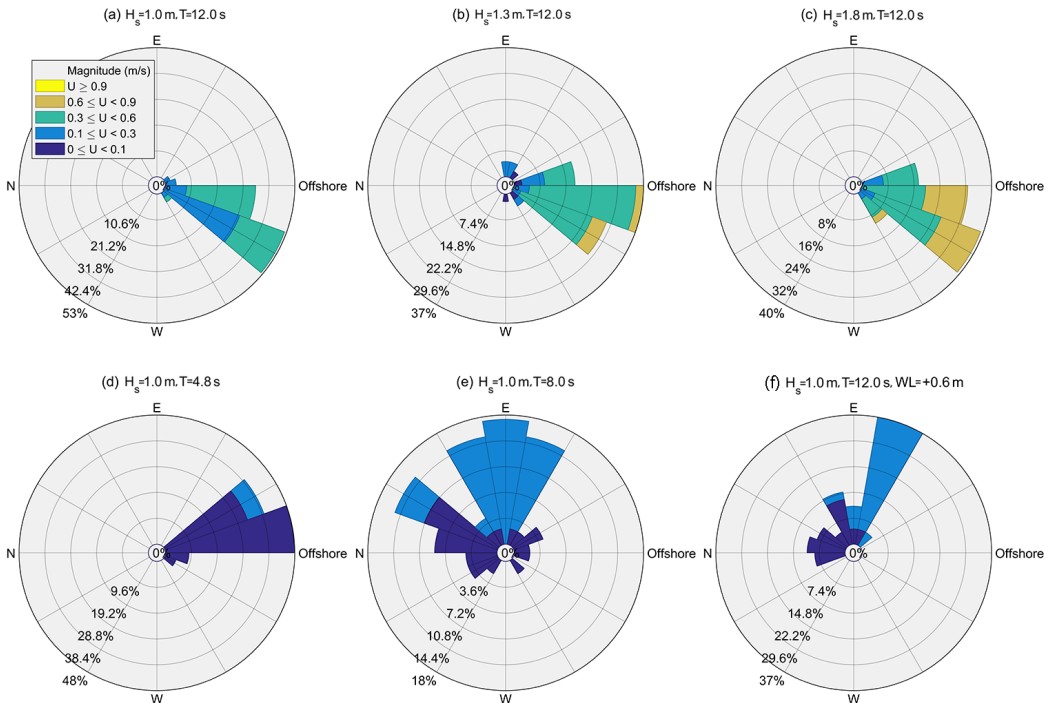

**Figure 5.** Rip current rose diagrams of rip currents forced by different wave conditions and tidal level at position $x$ = 730 m and $y$ = 120 m marked as Gauge A in Fig. 1f. Each spoke denotes the direction that the current flows to. The offshore direction is labeled in each diagram.



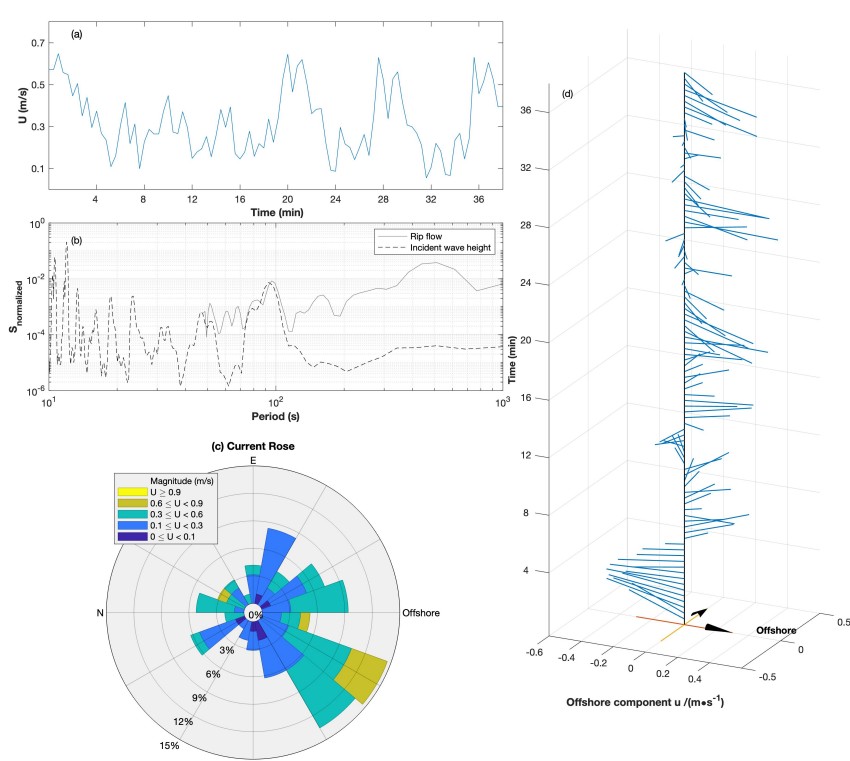

**Figure 6.** Analysis of rip flow variability in Gauge B: time series starting from 900 s (a); power spectra of rip flow and incident wave height (b); rose diagram and vector plot of Gauge B (c-d). The location of Gauge B is marked in Fig. 1f



**Figure 7.** Rip hazard maps for shore-normal incident waves of $H_s$ = 1.0 m, and $T_p$ = 12 s in Dadonghai Beach, using bathymetry on August 7, 2018 (a) and December 26, 2019 (b).

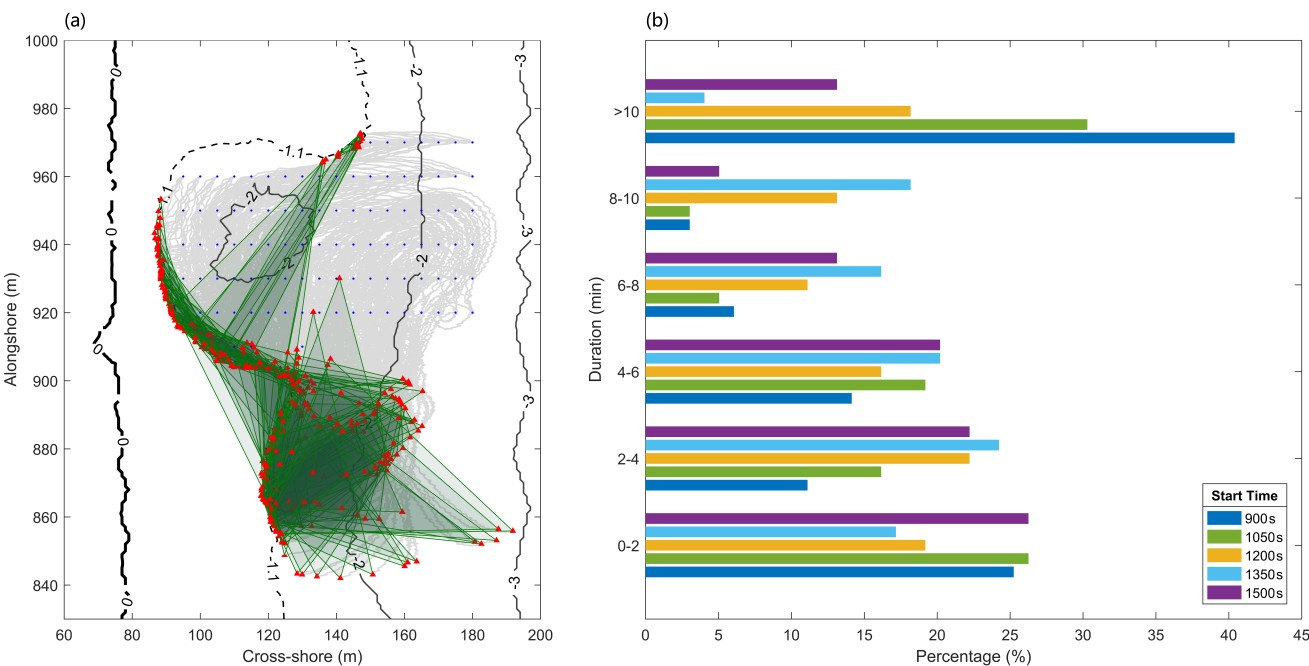

**Figure 8.** Lagrangian tracking of the virtual swimmers seeded at 5 different modeling times (with an interval of 150 s from 900 s to 1500 s) within Area 1 (a). Histograms of $t_{safe}$ given in percentages of swimmers who have reached safety at each $t_{safe}$ range, is shown in subplot (b). The initial seeding positions are marked by blue dots. At each position, the resulting 5 trajectories are plotted as light gray lines, with their ends (red triangles) connected by green lines. The shaded polygons are to highlight the variability of tracking outcomes. Surf-zone bathymetry is contoured as black lines.

**Stay afloat**

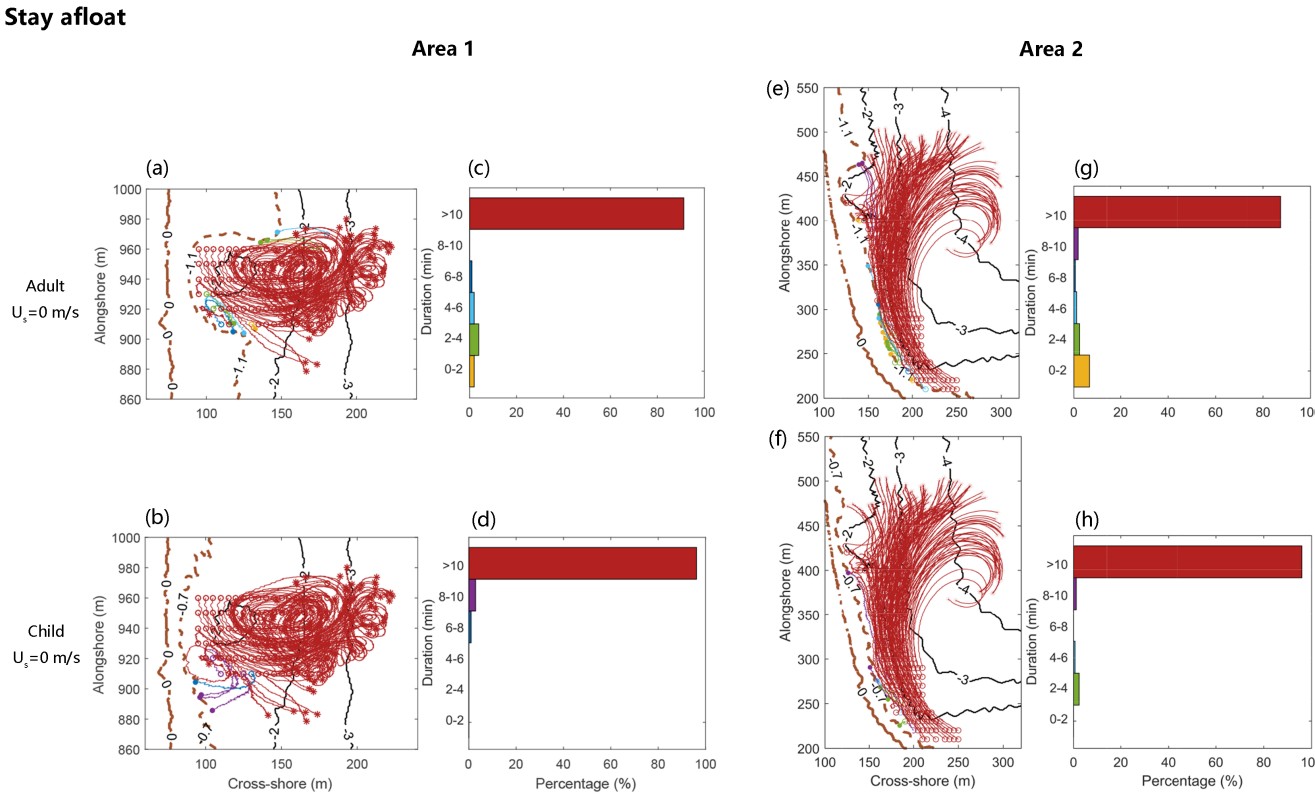

**Figure 9.** Swimmer tracking simulations for strategy of *stay afloat* within Area 1 (left panel), and Area 2 (right panel). Histograms of $t_{safe}$ give percentages of swimmers who have reached safety at each $t_{safe}$ range. The trajectories of swimmers have colors corresponding to bins in each histogram. Surf-zone bathymetry is contoured as black solid lines and brown dashed lines (0, 0.7 and 1.1 coutour lines).



## Swim onshore

**Area 1**

**Area 2**



**Figure 10.** Swimmer tracking simulations for strategy of *swimming Onshore* within Area 1 (left panel), and Area 2 (right panel). Histograms of $t_{safe}$ give percentages of swimmers who have reached safety at each $t_{safe}$ range. The trajectories of swimmers have colors corresponding to bins in each histogram. Surf-zone bathymetry is contoured as black solid lines and brown dashed lines (0, 0.7 and 1.1 coutour lines).



**Swim parallel
(Westward)**



**Figure 11.** Swimmer tracking simulations for strategy of *swimming westward parallel to shore* within Area 1 (left panel), and Area 2 (right panel). Histograms of $t_{safe}$ give percentages of swimmers who have reached safety at each $t_{safe}$ range. The trajectories of swimmers have colors corresponding to bins in each histogram. Surf-zone bathymetry is contoured as black solid lines and brown dashed lines (0, 0.7 and 1.1 coutour lines).

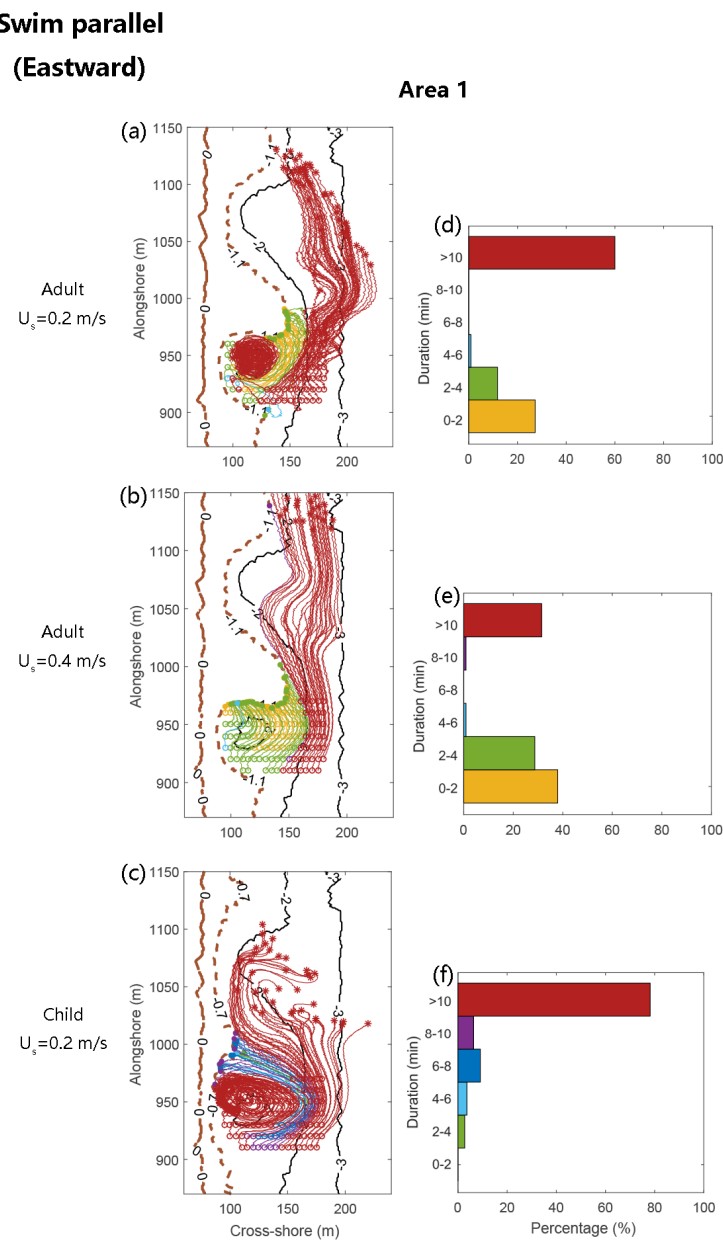

**Figure 12.** Swimmer tracking simulations for strategy of *swimming eastward parallel to shore* within Area 1. Histograms of $t_{safe}$ give percentages of swimmers who have reached safety at each $t_{safe}$ range. The trajectories of swimmers have colors corresponding to bins in each histogram. Surf-zone bathymetry is contoured as black solid lines and brown dashed lines (0, 0.7 and 1.1 coutour lines).





**Figure 13.** 10-min swimmer tracking simulations by instantaneous velocity (light gray line with start and end marked by open circles and solid triangles) and 24 s-averaged mean velocity (dark gray line), respectively. The ends of each pair of trajectories are connected by orange lines to denote the difference of tracking. Surf-zone bathymetry obtained in 2018 (top panel) and 2019 (lower panel) in Area 1 are used, which is overlapped by black contour lines. Histograms of $t_{safe}$ (right subplots) give percentages of swimmers who have reached safety at each $t_{safe}$ range. $U_{ins}$ and $U_{mean}$ denote the instantaneous and wave-averaged velocity, respectively.