# Peer review of "A wave-resolving modeling study of rip current variability, rip hazard, and swimmer escape strategies on an embayed beach with irregular rip channels"

_EGUsphere, 2023_

## Referee Comment (RC1)

[referee-annotated manuscript omitted]

---

## Author Comment (AC1)

**Authors' response to referee #1: egusphere-2023-122 "A wave-resolving modeling study of rip current variability, rip hazard, and swimmer escape strategies on an embayed beach with irregular rip channels"**

Dear Editor and Referees,

Thank you for your constructive comments on our manuscript entitled 'A wave-resolving modeling study of rip current variability, rip hazard, and swimmer escape strategies on an embayed beach with irregular rip channels'. We feel indebted to the reviewers for their time on this manuscript. We have incorporated the suggestions/comments in the revised manuscript. In the response below, all the comments and concerns are replied point by point.

Kind Regards,
Huaiwei Yang and Ye Yuan, on behalf of the co-authors.

―――――――――――――――

*The manuscript authored by Yuan et al studies rip current variability, real-time rip hazard identification, and the optimal swimmer escape strategies at Dadonghai Beach, China considering two bathymetric scenarios between 2018 and 2019.*
*Overall, the document is very well structured, details each section adequately, and is written in fluent English. General comments are indicated below and more detailed comments are in the attached document:*

*1. How accurate are the bathymetries derived from satellite imagery concerning measurements? Have any checks been made in the study case?*

**Response**: Local bathymetric patterns are critical to the modeling of nearshore circulation. However, in this study the detailed echosounder survey was not performed. Beach morphologies were inverted from two satellite images acquired from Google Earth Historical Database on August 7, 2018 and December 26, 2019 when sea state was calm. Using inverted depth has pros and cons. While accuracy of remote-sensed bathymetry sometimes is questionable, it is easily available by satellite and UAV, making it desirable for numerical forecasting of rip hazards.

The commonly-used techniques which allow indirect measurement of depth by satellite images are based on water color and wave kinematics. Colour-based methods are usually limited to non-turbid or non-breaking waters. The wave kinematics methods work well in turbid or optically deep water. A field trip of Dadonghai Beach was conducted on December 9, 2019 by SOA. During the survey, an UAV was released to observe beach morphology. As shown in Figure R1 and R2, well-developed, periodic crescent sandbars were recorded by the UAV, whose locations coincided with those appeared in the satellite image (Figure R2b). According to photos and public surveys of swimmers and coastal patrols,

water depth over the shore-connected sandbars varied approximately between 0.5 - 1.0 m, and 1.5 -

2.0 m over the rip channels. Floating separation ropes were deployed perpendicularly to the beach by coastal patrols to prevent bathers from swimming to deeper rip channels. The survey results were generally consistent with the inverted beach bathymetry. Though no precise bathymetric measurement was conducted by the authors, the beach morphological features could be reflected by the satellite images, which are thought to be adequate for models to grasp the main rip patterns.

[Figure]

Figure R1. Crescent sandbars recorded by the UAV during the rip-hazard field investigation in 2019 (Courtesy of National Marine Hazard Mitigation Service, China).

[Figure]

Figure R2. Crescent sandbars recorded in the satellite (b) and UAV (c) images, respectively. The satellite image was collected in October 2019 (Google Earth Historic Imagery), and the UAV image was collected during the rip-hazard field investigation at the same period (Courtesy of National Marine Hazard Mitigation Service, China).

**2. Is the beach in dynamic equilibrium, preserving sediment balance? Are the scenarios analyzed from the 2018 and 2019 satellite-derived bathymetries expected to be representative of all summers?**

**Response**: Dadonghai Beach is a typical medium-size embayment with headlands at both ends. It is classified as a typical headland-embayed beach (Wang et al., 2001). Based on the planform stability, Hsu et al. (2008) classified the headland bay beaches mainly as states of : (1) static equilibrium, (2) dynamic equilibrium, and (3) unstable . Beaches in a state of dynamic equilibrium, waves breaks at an angle to the shoreline at headlands, generating longshore currents capable of transporting sediment shoreward. Beach morphology is constantly adjusting to changes or variations in forcings in different temporal scales, i.e., short-term changes, seasonal variability. It is expected that embayed beach shorelines tend to be relatively stable over long timescales (Silvester and Hsu, 1997; Daly et al., 2015).

At Dadonghai Beach, incoming waves breaks at headlands, which contributes to the continuous sediment supply to the bay interior. The rhythmic sandbars are well developed along the beach, and its periodic evolvement is subject to storm event-driven or seasonal changes (mainly moonsoon seasons and typhoon seasons). According to the historical satellite images (Figure R3), beach morphology at the Dadonghai Beach was evolving periodically, and exhibited different morphological stages, from a nearly straight shore-parallel sandbar approximately 80-100 m seaward from the shoreline, to crescentic or welded bars, and then to transverse bars with incised, narrow rip channels (Figure R3). Sandbar migration can also be observed due to longshore currents. These stages are typical equilibrium beach responses at wave-dominated beaches (Castelle & Masselink, 2023).

Two satellite images of the Dadonghai Beach on August 7, 2018 and December 26, 2019 are used in the study, representing summer and winter scenarios. In summer, frequent typhoon activities in the South China Sea (normally from July to October) generate long-period swells periodically received by the Dadonghai; while in winter the NE and E monsoon prevails in the Northern SCS and brings swells into the embayment. By examining historical images in Figure R3, however we cannot find obvious seasonal variations in beach morphology at Dadonghai Beach. Detailed study on seasonal modulation of beach morphodynamics need to be carried out in future.

[Figure]

Figure R3. Variations of beach morphology at Dadonghai Beach from 2017 to 2022.

*3. The model used to obtain the currents has been validated with field measurements? this is essential to validate the directions and magnitudes of the velocities.*

**Response**: Although FUNWAVE-TVD and similar Boussinesq-type wave model (i.e., FunwaveC) has been widely proved to be an effective tool to study wave-induced nearshore circulations and dispersion (Chen, et al., 1999, 2003; Geiman et al., 2013; Feddersen, 2014), in this study field measurements of rip currents at Dahonghai Beach has not been conducted to validate the model. Lack of field observations considerably limits our further analysis on rip variability, and their response to beach morphology and wave dynamics.

As shown in Figure R4, sediment plumes, or 'rip heads' are clearly visible from the CCTV of the Dadonghai Beach, which are indicative of mega rip modeled by the FUNWAVE-TVD in the study. As described in Section 4.1 of the manuscript, the incident waves shoal and break at the west headland and coral reefs immediately after entering the embayment, and in turn produce strong longshore currents. The longshore currents then deflect to the offshore direction, forming a persistent mega rip. The FUNWAVE can capture this feature properly, which lends us some confidence on FUNWAVE's modeling results.

To remind readers of the absence of observational data, we made some revisions within the manuscript (i.e., emphasizing lack of observations and limitation of the study in the Conclusion; Line * in Abstract, 'rip currents fluctuate' => 'the modeled rip currents fluctuate').

[Figure]

Figure R4. Sediment plumes observed at the westmost corner of the Dadonghai Beach (denoted by black arrows; this image is from Li et al., 2020), which corresponds to the mega rip modeled by FUNWAVE-TVD in the study.

*4. Where is the wave point located? It is essential to know its location in order to describe the wave conditions. This point should be indicated on the location map in Figure 1. Why not evaluate the entire wave time series considering the 30 years characterizing summers and winters and then specifically 2017-2018?*

**Response:** The wave point is located at 18.2N, 109.5E, which is marked as a yellow triangle in Figure 1a. Wave rose diagram of 30-year-long time series of wave hindcast off the Dadonghai Bay is included in Figure 2, which is also shown in Figure R5 in this document. *Section 2.2 Wave conditions* has been reformulated as follows:

Based on an analysis of 30-year wave hindcast dataset developed by National Marine Environmental Forecasting Center of China (NMEFC), wave conditions immediately off the Dadonghai Bay are assessed. The wave point for analysis is located at 109.5 E, 18.2 N, which is denoted as a white triangle in Fig. 1a. According to wave rose diagram of 30-year-long wave hindcast (Fig. 2a), generally the Dadonghai Bay receives waves at 2 prevailing directions, including powerful typhoon swells mainly from the south, and monsoon swells from the southeast. The 1-day moving average on the hourly wave hindcast in 2018 is performed and shown in Fig. 2d for further analysis. Waves during the summer months are relatively more energetic than in winter immediately off the Dadonghai Bay, which is interspersed with high-energy events associated with typhoon activities in the northern South China Sea (SCS). Typhoons can send long-period swells to the Dadonghai Bay. Wave-buoy observations along the slope of the Northern SCS demonstrate that while peak wave period during the winter monsoon is between 4 - 8 second, it is capable to reach up to 10 - 14 second during the passage of the tropical cyclones in summer (Xu et al., 2017; Tian et al., 2020). As shown in Fig. 2c, these storm waves arrive over a wide spread of directions from SE to SW. As typhoons enter the northern SCS and move west, the directions of waves receive by the Dadonghai Bay vary over time. Two prominent peaks in July and August are explained by two typhoons moving westward through the Northern SCS in 2018 summer, leading to elevated significant wave height of 1.5 - 2.5 and 2.0 - 4.0 m (without 1-day moving average), respectively. Strong typhoon swells persist for a week. During the winter months, the prevailing winter monsoon sometimes produce northeast wind waves in the Northern SCS with significant wave height of more than 3 m. However, as the monsoon swells propagate into the bay from the open sea, the waves get weaker in wave height and are diffracted to the southeast due to shelter of the peninsula to the east (Fig. 2b). The hourly significant wave height off the Dadonghai Bay generally varies below 1.0 m, and can reach up to 2.0 m with the outbreak of monsoon.

Accordingly, various wave conditions representative of winter monsoon swells and summer typhoon swells are used in the following modeling studies of rip variability and hazards.

[Figure]

Figure R5. Wave climate in study site. Subplot (a) is wave rose diagram of 30-year-long time series of wave hindcast at a wave point immediately off the Dadonghai Bay (109.5 E, 18.2 N); (b-c) are wave rose diagrams for winter monsoon and summer typhoon seasons in 2018, respectively; (d) is 1 day-averaged time series of significant wave height in 2018.

***Details:***

***1. The abstract is quite long and it is preferable to be more concise in order to attract the readers' attention.***

**Response:** Accepted. The abstract has been shortened. Please refer to the revised manuscript.

***2. Paragraph between Line 108 and 112 is not a description of the bathymetry. It should be included as another subsection describing sea levels.***

**Response:** Accepted. Indeed it is inappropriate to include this paragraph in Section 2.1. The first sentence of the paragraph presents the information about tidal range of the Dadonghai Beach, and the rest of the paragraph is about bedrock morphology at both ends of the bay. Instead of starting another subsection about tidal levels, we removed this paragraph from Section 2.1 and merged it into the opening paragraph of Section 2. Please refer to revised manuscript.

***3. Table 1: How were these 9 tests defined? Are they the most frequent characteristics of summers? If the selected tests are representative of summer wave conditions, it would be appropriate to deepen this analysis in section 2.2 by considering the complete wave time series.***

**Response:** Accepted. *Section 2.2* has been reformulated. Wave conditions given in 9 tests are representative of winter monsoon swells and summer typhoon swells. Among these wave conditions, $T_p = 4.8$ s and 8.0 s are representative of short-period monsoon swells in winter, and $T_p = 12.0$ s is

representative of long-period typhoon-induced swells in summer. Wave incident angles of 5° and 20° is also adopted to represent oblique SE or SW swells. The 2$^{nd}$ paragraph of *Section 3.2* has been slightly revised.

***6. Line 227 - 235: Do these diagrams consider the results of all simulations in all tests?***

**Response:** Statistical analysis of Test 1, 7 and 8 is not included in Figure 5. The generated rip currents are weak in Test 1 with wave forcing of $H_s$=0.7 m and $T_p$=12 s. Test 7 and 8 are aimed to test the effect of wave incident angle on rip currents. The modeling results show that nearshore circulation is dominated by persistent longshore currents. We made a short statement in the caption of Figure 5 to explain that T1 and T7-8 were not included.

*Editorial errors:*

*Line 42: exist => exit;*
*Line50: determine => determines;*
*Line 53: hazard => hazards;*
*Line 54: How => how, level => level of;*
*Line 94: beach => Beach;*
*Line 110: coastal coral reef => a coastal coral reef;*
*Line 112: been => be;*
*Line 121: add 'the' before NMEFC;*
*Line 124: averaging => average;*
*Line 140: In our simulation => In this study;*

*Table 1: defining λ;*

*Line 155: are => is;*
*Line 157: add 'the' before 'hydrodynamic response';*
*Line 166: We define rip currents => In this study the rip currents are defined as...;*
*Line 171: add 'the' before 'greatest danger';*
*Line 172: location => locations;*
*Line 176: add 'the' before 'moverment';*
*Line 184: add 'a' before 'safe state';*
*Line 195: move => moves;*
*Line 204: deflect => deflects*
*Line 246: an => a;*
*Line 285: remove space;*
*Line 298: child => a child;*
*Line 312: to Child => for children;*
*Line 342 & 353: AREA => Area;*
*Line 401: Floating => floating;*
**Response:** All the editorial errors have been corrected in the revised manuscript.

*Figure improvement:*
*Figure1: Indicate geographical north orientation; Define zoom frame panels b and d; Include the location of the wave series point.*

**Response:** Symbols for geographical north orientation and wave point have been added. However, scale in Figure 1a is too large to fit the zoom frame for panels b and d.

*Figure 2: Preferable to indicate time in actual dates.*

**Response:** Accepted. The julian day is replaced with month names.

*Figure 3 & 4: (1)The "s" in Hs does not need to be subscripted; (2)It is recommended to superimpose guide vectors over the currents to highlight the predominance; (3)Why is the wave direction specified only in the last 2 panels and not in the rest? (4)Panels e to i would correspond to tests 1 to 8. Where would the result of test 9 be?*

**Response:** (1) The bulk parameter Peak Significant Wave Height is denoted with $H_s$ in the study. We also checked with the related literature. Most of them subscript letter 's' to denote significant wave height. (2) There were several versions of Figure 3&4 before we submitted the manuscript. Subplots with guided vectors was one of them. However, each row of subplots in the figure use different length scales of vectors, and labeling them individually may cause confusion. Finally, we decided to use shaded contours overlapped with vectors to highlight the surf-zone flows. Here the vectors are just for illustration.(3) All the parameters defining wave conditions are listed in each subplot. (4) Test 9 shows how variation in tidal level affect rip currents. The result is illustrated in Figure 5f. As the nearshore circulation is too weak in Test 9, we haven't included it in Figure 3&4.

*Figure 7: Specify which is this test indicating wave direction and tide level.*

**Response:** Accepted. Figure caption has been revised.

**Reference:**

(1) Castelle B., G. Masselink. Morphodynamics of wave-dominated beaches. Cambridge Prisms: Coastal Futures, 2023, 1, pp.e1. ff10.1017/cft.2022.2ff. ffhal-03830565

(2) Daly,C. J., C. Winter, Karin R. Bryan, On the morphological development of embayed beaches, Geomorphology, 248, 252-263, 2015.

(3) Chen, Q., Dalrymple, R. A., Kirby, J. T., Kennedy, A. and Haller, M. C., 1999, ``Boussinesq modeling of a rip current system'' , Journal of Geophysical Research, 104, 20,617 - 20, 637.

(4) Chen, Q., Kirby, J. T., Dalrymple, R. A., Shi, F., & Thornton, E. B. (2003). Boussinesq modeling of longshore currents. Journal of Geophysical Research, 108(11), 3362.

(5) Feddersen, F., 2014: The Generation of Surfzone Eddies in a Strong Alongshore Current. J. Phys. Oceanogr., 44, 600–617, https://doi.org/10.1175/JPO-D-13-051.1.

(6) Geiman, J. D., and J. T. Kirby, 2013: Unforced Oscillation of Rip-Current Vortex Cells. J. Phys. Oceanogr., 43, 477–497, https://doi.org/10.1175/JPO-D-11-0164.1.

(7) Hsu, J. R-C.; Evans, C. Parabolic bay shapes and applications. In: Proceedings of Institution of Civil Engineers - Part 2. London: Thomas Telford, 1989.

(8) Silvester, R., John R. C. Hsu. Coastal Stabilization. World Scientific, 1997

(8) Tian, D.; Zhang, H.; Zhang, W.; Zhou, F.; Sun, X.; Zhou, Y.; Ke, D. Wave Glider Observations of Surface Waves During Three Tropical Cyclones in the South China Sea. Water 2020, 12, 1331. https://doi.org/10.3390/w12051331

(9) Xu, Y., H. He, J. Song, Y. Hou, and F. Li, 2017: Observations and Modeling of Typhoon Waves in the South China Sea. J. Phys. Oceanogr., 47, 1307–1324, https://doi.org/10.1175/JPO-D-16-0174.1.

(10) Wang, Ying & Martini, I. Peter & Zhu, Dakui & Zhang, Yongzhan & Tang, Wenwu. (2001). Coastal plain evolution in southern Hainan Island, China. Chinese Science Bulletin. 46. 90-96. 10.1007/BF03187244.

---

## Author Comment (AC2)

**Authors' response to referee #2: egusphere-2023-122 "A wave-resolving modeling study of rip current variability, rip hazard, and swimmer escape strategies on an embayed beach with irregular rip channels"**

Dear Editor and Referees,

Thank you for your constructive comments on our manuscript entitled 'A wave-resolving modeling study of rip current variability, rip hazard, and swimmer escape strategies on an embayed beach with irregular rip channels'. We feel indebted to the reviewers for their time on this manuscript. We have incorporated the suggestions/comments in the revised manuscript. In the response below, all the comments and concerns are replied point by point.

Kind Regards,
Huaiwei Yang and Ye Yuan, on behalf of the co-authors.
* * *
*The paper presents a Boussinesq model study of rip current variability and the evaluation of hazards induced by rip currents. The study showcases the trajectory tracking method used in an embayed beach area for swimmer escape strategies. It is an interesting study.*

*Because the first author was the developer of FUNWAVE-GPU, I don't have many concerns about modeling details. However, I feel it may need logical consistency across the entire article, from the introduction to the conclusions. The authors expressed the importance of a wave-resolving model for such a trajectory-tracking study due to the random and dynamic nature of rip currents. But the paper concluded that the results from the Boussinesq model are comparable to that from the wave-averaged method. If the swimmer escape strategies are the main objective of the study, using the expensive Boussinesq model seems to be an overkill. For this reason, I suggest that the authors may emphasize more about the effects of IG and VLF bands because a regular wave averaged model cannot predict IG motions without using a non-stationary wave condition. The VLF motions in the model results are interesting. It is good to check if those are the shear wave mode as said or long gravity wave oscillations. A plot of a wavenumber spectrum would be helpful.*

*In general, the paper was well-written and easy to follow. It should be published after addressing the issue mentioned above.*

**Response:** Thanks for the insightful comments.
First of all, we would like to clarify that the 'wave-averaged velocity' in the manuscript (i.e., Abstract, Section 5.2) is computed by averaging phase-resolving velocities in two wave periods (24 seconds) in FUNWAVE-TVD, rather than the velocity yield by other short-wave-averaged models such as XBeach-SB (XBeach-SurfBeat). Both the wave-resolving or wave-averaged tracking in the study all retained the non-stationary IG or VLF motions, thus presenting similar trajectories of virtual swimmers. We would like to reiterate here that the wave-resolving tracking means using instantaneous, random velocities at each timestep (~0.04 s) to tracer swimmers, while wave-averaged tracking uses 24 s-averaged mean flow velocity to tracer swimmers.

Previous beach-safety studies (McCarroll et al., 2015; Castelle et al., 2016) use XBeach-SB to study rip dynamics and Lagrangian tracking of virtual swimmers. XBeach has 2 different dynamic frameworks inside, namely 'Surf Beat (SB)' and 'Non-hydrostatic (NH)'. Though XBeach-SB is a phase-averaged surfzone model, it offers an additional advantage compared to other phase-averaged models in its ability to resolve wave-group generated infragravity (IG) motions, vortical currents at VLF timescales. XBeach-SB uses the 2D wave spectra at its offshore boundary to calculate the wave group envelope, and then solves the variation of short-waves envelope (wave height) on the scale of wave groups. This variation in turn drives IG waves within the depth-integrated hydrostatic solver, approximating transient behaviors and swash dynamics in the nearshore zone. The non-hydrostatic module (XBeach-NH) extends XBeach's capability to wave-resolving modeling of non-linear waves, wave-current interaction and wave breaking in the surf zone. Both Boussinesq-type wave model FUNWAVE and XBeach-NH are phase-resolving. The fundamental difference between them is that FUNWAVE is a depth-integrated model, and relies on higher-order derivative terms to improve the frequency dispersion, while it is improved in NH models by including vertical layers.

There was a comparison between power spectra of incoming waves and resulting flow fluctuations at a given point in Figure 6b. We have also set a list of output points along the beach, whose power spectra were not shown in the manuscript. The spectra demonstrated that the wave-group-forced IG waves were persistent at all points, though their amplitudes varied from point to point; while Very low-frequency [VLF, $O(10\ min)$] motions only existed at several points where rip flows pulsated in amplitude and directions. The absence of spectral peak of incoming waves at VLF band suggested that the VLF motions were not forced by the long-period component of gravity waves. Therefore, we concluded that the VLF motions were usually related to the formation of vortex due to local morphology. The detailed analysis of generation mechanism of VLF motions was not available in the manuscript due to lack of in-situ instrumental observations.

To avoid misunderstanding, we have made several revisions in the manuscript: (1) Abstract was shortened by removing the statement 'Virtual trajectories yielded by the wave-resolving and wave-averaged velocities are generally consistent with each other'; (2) The definitions of wave-resolving and wave-averaged tracking are explained in Section 5.2; (3) Similar statements in Conclusion were also revised.

***Details:***

***Figure 1(f), please provide the vertical datum for 2018 and 2019 measurements. Provide an accuracy estimate for the satellite-derived bathymetric data.***

**Response:**
(1) Two satellite images were acquired from Google Earth Historical Database on August 7, 2018 and December 26, 2019 when sea state was calm. However, the exact collecting time of the images was not clear to us. Therefore, the tidal levels when the satellite images were acquired could not be precisely determined from the nearly tidal gauge. According to the tidal records of nearly Sanya tidal gauge, Dadonghai Beach has a micro-tidal range, and tidal ranges on August 7, 2018 and December 26, 2019 were approximately 0.97 m and 1.62 m. The collecting time of the images were roughly estimated by the shadows of the tall buildings along the beach. We speculated that the picturing time on August 7, 2018 was around early morning when the projected shadows were elongated and on the west side of the building; while on December 26, 2019 it was at the mid-day. According to this information, the tidal

levels were around mid-to-high tide and mid-tide, respectively. The computation grids were constructed by combining inversion of satellite images along the beach and nautical chart within the Dadonghai Bay. The combined bathymetry were converted to the mean tidal level (MTL) based on the datum at Sanya Tidal gauge nearby.

(2) Local bathymetric patterns are critical to the modeling of nearshore circulation. However, in this study the detailed echosounder survey was not performed. Beach morphologies used in the study were inverted from true-color satellite images. The commonly-used techniques which allow indirect measurement of depth by satellite images are based on water color and wave kinematics. Colour-based methods are usually limited to non-turbid or non-breaking waters. The wave kinematics methods work well in turbid or optically deep water. A field trip of Dadonghai Beach was conducted on December 9, 2019 by SOA. During the survey, an UAV was released to observe beach morphology. As shown in Figure R1c, well-developed, periodic crescent sandbars were recorded by the UAV, whose locations coincided with those appeared in the satellite image (Figure R1b). According to photos and public surveys of swimmers and coastal patrols, water depth over the shore-connected sandbars varied approximately between $0.5-1.0$ m, and $1.5-2.0$ m over the rip channels. Floating separation ropes were deployed perpendicularly to the beach by coastal patrols to prevent bathers from swimming to deeper rip channels. The survey results were generally consistent with the inverted beach bathymetry. As a summary, though no precise bathymetric measurement was conducted by the authors, the beach morphological patterns could be reflected by the satellite images, and the inverted depth are adequate for modeling study.

[Figure]

Figure R1. Crescent sandbars recorded in the satellite (b) and UAV (c) images, respectively. The satellite image was collected on December 13, 2019 (Google Earth Historic Imagery), and the UAV image was collected during the rip-hazard field investigation at the same period on December 9, 2019 (Courtesy of National Marine Hazard Mitigation Service, China).

*How does a periodic boundary condition set up in such a bay-like domain?*

**Response:** In the study a south open boundary is placed in the grid, and there is no east-west open boundaries. Therefore, lateral periodic boundary condition is not necessary. The expression in Line 116 is revised accordingly as follows.
'The rotation is necessary for FUNWAVE to apply irregular wave maker and periodic boundary' =>
'The rotation is necessary for FUNWAVE to apply internal wavemaker'.

*Line 112, both ends is exposed, grammar*

**Response:** 'Both ends is exposed' => 'Both ends are exposed'.

*Line 139, FUNWAVE-GPU add the reference here*

**Response:** The reference *Yuan et al., 2020* has been added here.

*Line 142, CFL, add the complete terminology*

**Response:** 'CFL' is replaced with 'Courant–Friedrichs–Lewy (CFL)' when first appeared.

*Line 144, friction coefficient, need to clarify bottom friction form, manning formula? or provide a reference*

**Response**: A constant bottom drag coefficient of 0.0025 in the quadratic friction formula was applied. We added a reference of Zhang et al., 2022 in the manuscript.

> Zhang, Y., Shi, F., Kirby, J. T., & Feng, X. (2022). Phase-resolved modeling of wave interference and its effects on nearshore circulation in a large ebb shoal-beach system. Journal of Geophysical Research: Oceans, 127, e2022JC018623. https://doi.org/10.1029/2022JC018623

*Section 3.3. It's interesting to make a definition for the hazard levels. Any reference for this definition, or just created by the authors?*

**Response:** The rip hazard levels are created by the authors. It is preferable to quantify and visualize the rip hazard by simple and effective indexes or guidelines for operational centers or coastal patrol. Thus, we proposed an index table for rip hazard levels by combining rip strength and duration.

*Line 181, arbitrary factor of 0.8. need a sensitivity test on this number*

**Response:** We have made a sensitivity study in the arbitrary factor (0.8) in Equation 1 ($u_{tracking} = 0.8u_w + U_s$, where $u_{tracking}$ is Lagrangian tracking velocity, $u_w$ is instantaneous wave-induced rip flow, and $U_s$ is swimming velocity) defining how well a tracer (virtual swimmer) drifts with the ambient flow. In *Section 'Discussion'* of the revised manuscript, we have included results of the sensitivity study by varying the factor from 0.4 to 1.0. An brief analysis is included here.

This arbitrary factor is defined as the floating factor varying from 0.4 to 1.0 with an interval of 0.2. As shown in Figure R2, We mainly focus on the swimmers that do not reach safety after 10-min swimming onshore (trajectories with red color). In the case that the floating factor of 0.4 is specified, almost entire swimmers can get safe by swimming onshore with an average swimming velocity of 0.2 m/s, even for virtual swimmers that are deployed in outer surf zone. However, at the opposite extreme, more than 40% of swimmers are exhausted in the rip eddy and fail to reach safe areas when the floating factor of 1.0 is set in the model.

The sensitivity study suggests that the value of floating factor is crucial to the tracking results of virtual swimmers, which in turn influences swimmer escape strategies. The factor should be calibrated in the further field studies.

[Figure]

Figure R2. Sensitivity study of floating factor in Equation 1. Histograms of $t_{safe}$ give percentages of swimmers who have reached safety at each $t_{safe}$ range.The factor varies from 0.4 to 1.0 with an interval of 0.2 to define how well swimmers float with the ambient wave-induced flows (a-d). Assigning 0.4 means that swimmers are only slightly affected by surf-zone flows, and assigning 1.0 means that swimmers float with the ambient water perfectly. In this case, virtual swimmers have a constant onshore swimming velocity of 0.2 m/s.

**Reference:**

(1) Castelle, B., McCarroll, R., Brander, R., T., S., and B., D.: Modelling the alongshore variability of optimum rip current escape strategies on a multiple rip-channelled beach, Nat Hazards, 81, 663–686, https://doi.org/10.1007/s11069-015-2101-3, 2016.

(2) McCarroll, R. J., Castelle, B., Brander, R., and T., S.: Modelling rip current flow and bather escape strategies across a transverse bar and rip channel morphology, Geomorphology, 246, 502–518, https://doi.org/10.1016/j.geomorph.2015.06.041, 2015.

(3) Salatin, Reza & Chen, Qin & Bak, A. & Shi, Fengyan & Brandt, Steven. (2021). Effects of Wave Coherence on Longshore Variability of Nearshore Wave Processes. Journal of Geophysical Research: Oceans. 126. 10.1029/2021JC017641.

(4) Zhang, yu & Shi, Fengyan & Kirby, James & Feng, Xi. (2022). Phase-Resolved Modeling of Wave Interference and Its Effects on Nearshore Circulation in a Large Ebb Shoal-Beach System. Journal of Geophysical Research: Oceans. 127. 10.1029/2022JC018623.

---

## Author Response (AR1)

**Authors' response to the Editor and Referees**
**on**
**egusphere-2023-122 "A wave-resolving modeling study of rip current variability, rip hazard, and swimmer escape strategies on an embayed beach with irregular rip channels"**

Dear Prof. Mauricio Gonzalez and Referees,

Thank you for your constructive comments on our manuscript entitled 'A wave-resolving modeling study of rip current variability, rip hazard, and swimmer escape strategies on an embayed beach with irregular rip channels'. We feel indebted to the reviewers for their time on this manuscript. In the response below, all the comments and concerns are replied point by point. The revised manuscript with tracked changes is provided separately.

The ***italic & bold*** texts denote the referees' comments, and the blue texts denote the revised sentences or paragraphs. The line numbers mentioned in the responses are based on the revised manuscript with tracked changes.

Kind Regards,
Huaiwei Yang and Ye Yuan, on behalf of the co-authors.

**Response to referee #1:**

*The manuscript authored by Yuan et al studies rip current variability, real-time rip hazard identification, and the optimal swimmer escape strategies at Dadonghai Beach, China considering two bathymetric scenarios between 2018 and 2019.*
*Overall, the document is very well structured, details each section adequately, and is written in fluent English. General comments are indicated below and more detailed comments are in the attached document:*

*1. How accurate are the bathymetries derived from satellite imagery concerning measurements? Have any checks been made in the study case?*

**Response**: Local bathymetric patterns are critical to the modeling of nearshore circulation. However, in this study the detailed echosounder survey covering the Dadonghai Bay was not performed. Beach bathymetries were inverted from two satellite images acquired from Google Earth Historical Database on August 7, 2018 and December 26, 2019 when sea state was calm. The commonly-used techniques which allow indirect measurement of depth by satellite images are based on water color and wave kinematics. Colour-based methods are usually limited to non-turbid or non-breaking waters. The wave kinematics methods work well in turbid or optically deep water. While remote-sensed bathymetry is not as accurate as in-situ measurement, it is easily available by satellite and UAV, making it desirable for operational forecasting of rip hazards.

The depth data for the inversion were from two sources, including nautical charts and field survey. For main body of the Dadonghai Bay, large-scale nautical chart (1:25,000, purchased from China

Navigation Press) was interpolated and converted to the mean tidal level (MTL) based on the datum at Sanya Tidal gauge nearby, which was then correlated against the corresponding color pixels.

Surf-zone bathymetry is constantly changing in multiple temporal scales. For surf zone of the Dadonghai Beach that was not covered by the nautical chart, we relied on some in-situ photos, UAV images and public surveys collected from a field trip of Dadonghai Beach conducted in October 2019 by National Marine Hazard Mitigation Service of State Oceanic Administration (NMHMS). These information and images were combined to derive appropriate color-depth relation in the surf zone. During the survey, an UAV was released to observe beach morphology. As shown in Figure R1 and R2c (courtesy of NMHMS), well-developed, periodic crescent sandbars were recorded by the UAV, whose locations coincided with those appeared in the satellite image (Figure R2b). According to photos and public surveys of swimmers and coastal patrols collected during the field trip, water depth varied

approximately between 0.5‒1.0 m over the shore-connected sandbars, and 1.5‒2.0 m over the rip

channels. Floating separation ropes were deployed perpendicularly to the beach by coastal patrols to prevent bathers from swimming to deeper rip channels. Though no precise bathymetric measurement was conducted in the study, the major features of beach morphological could be reflected by the satellite images, which are thought to be adequate for models to grasp the main rip patterns.

In summary, we fully understand that the lack of accurate measurement of bathymetry may limit the credibility of the current modeling study. In the revised manuscript, we thus re-structured 2$^{nd}$ paragraph of *Section 2.1 Surf-zone bathymetry* (Line 118-131) as follows.

"Compared to extensive sonar or in-situ measurements of depth, shallow bathymetry can be fast and cost-effectively evaluated by remote-sensing images. In this study, by establishing a site-specific linear relationship between pixel colors and depths, nearshore bathymetry at Dadonghai was mapped and interpolated to 1-meter resolution. Although this inversion may not produce bathymetry as accurate as other approaches, it can be operationalized for rip hazard forecast in future owing to its simplicity to locate sandbars and shoals, as well as availability of satellite imagery (Radermacher et al., 2018). It should be noted that the in-situ echosounder survey was not performed in August 2018 and December 2019 for this study. For main body of the Dadonghai Bay, the depth data for the color-depth correlation was from nautical chart published by China Navigation Press with a scale of 1:25000, which was converted to the mean tidal level (MTL) based on the datum at Sanya Tidal gauge nearby. Surf-zone bathymetry is constantly changing in multiple temporal scales. For the surf zone that is not covered by the nautical chart, only very limited field data collected in October 2019 was available to derive the color-depth relation along the surf zone. The derived bathymetry was then rotated 90 degrees to align the shoreline with the vertical axis. The rotation is necessary for FUNWAVE to apply irregular wave maker."

Besides, we also included our thanks to NMHMS/SOA for providing some field data in the section of *Acknowledgement* (Line 478-479).

[Figure]

Figure R1. Crescent sandbars recorded by the UAV during the rip-hazard field investigation in 2019 (Courtesy of National Marine Hazard Mitigation Service, China).

[Figure]

Figure R2. Crescent sandbars recorded in the satellite (b) and UAV (c) images, respectively. The satellite image was collected in October 2019 (Google Earth Historic Imagery), and the UAV image was collected during the rip-hazard field investigation at the same period (Courtesy of National Marine Hazard Mitigation Service, China).

**2. Is the beach in dynamic equilibrium, preserving sediment balance? Are the scenarios analyzed from the 2018 and 2019 satellite-derived bathymetries expected to be representative of all summers?**

Dadonghai Beach is a typical medium-size embayment with headlands at both ends. It is classified as a typical headland-embayed beach (Wang et al., 2001). Based on the planform stability, Hsu et al. (2008) classified the headland bay beaches mainly as states of : (1) static equilibrium, (2) dynamic equilibrium, and (3) unstable . Beaches in a state of dynamic equilibrium, waves breaks at an angle to the shoreline at headlands, generating longshore currents capable of transporting sediment shoreward. Beach morphology is constantly adjusting to changes or variations in forcings in different temporal scales. It is expected that embayed beach shorelines tend to be relatively stable over long timescales (Silvester and Hsu, 1997; Daly et al., 2015).

Dadonghai Beach has been long known as one of the most famous bathing beaches in the South China for several decades. At Dadonghai Beach, incoming waves breaks at headlands, which contributes to the continuous sediment supply to the bay interior. The rhythmic sandbars are well developed along the beach, and its periodic evolvement is subject to storm event-driven or seasonal changes (mainly moonsoon seasons and typhoon seasons). According to the historical satellite images (Figure R3), beach morphology at the Dadonghai Beach was evolving periodically, and exhibited different morphological stages, from a nearly straight shore-parallel sandbar approximately 80-100 m seaward from the shoreline, to crescentic or welded bars, and then to transverse bars with incised, narrow rip channels (Figure R3). Sandbar migration can also be observed due to longshore currents. These stages are typical equilibrium beach responses at wave-dominated beaches (Castelle & Masselink, 2023).

Two satellite images of the Dadonghai Beach on August 7, 2018 and December 26, 2019 are used in the study, representing summer and winter scenarios. In summer, frequent typhoon activities in the South China Sea (normally from July to October) generate long-period swells periodically received by the Dadonghai; while in winter the NE and E monsoon prevails in the Northern SCS and brings swells into the embayment. By examining historical images at Google Earth Database, beach stages at Dadonghai Beach changes constantly with clear periodic cycles, however we cannot find obvious seasonal variations between summer and winter seasons.

[Figure]

Figure R3. Variations of beach morphology at Dadonghai Beach from 2017 to 2022.

***3. The model used to obtain the currents has been validated with field measurements? this is essential to validate the directions and magnitudes of the velocities.***

**Response**: Although FUNWAVE-TVD and similar Boussinesq-type wave model (i.e., FunwaveC, Coulwave) has been widely proved to be an effective tool to study wave-induced nearshore circulations and dispersion (Chen, et al., 1999, 2003; Geiman et al., 2013; Feddersen, 2014), in this study field measurements of rip currents at Dahonghai Beach has not been conducted to validate the model. Lack of field observations considerably limits our further analysis on rip variability, and their response to beach morphology and wave dynamics.

As shown in Figure R4, sediment plumes, or 'rip heads' are clearly visible from the CCTV of the Dadonghai Beach, which are indicative of mega rip modeled by the FUNWAVE-TVD in the study. As described in *Section 4.1* of the manuscript, the incident waves shoal and break at the west headland and coral reefs immediately after entering the embayment, and in turn produce strong longshore currents. The longshore currents then deflect to the offshore direction, forming a persistent mega rip. The FUNWAVE can capture this feature properly, which lends us some confidence on FUNWAVE's modeling results.

[Figure]

Figure R4. Sediment plumes observed at the westmost corner of the Dadonghai Beach (denoted by black arrows; this image is from Li et al., 2016; Wang et al., 2018), whose location corresponds to the mega rip modeled by FUNWAVE-TVD in the study.

To remind readers of limitation of this study, we made some revisions within the manuscript as follows.

(1) Abstract Line 9: "rip currents fluctuate" => "the modeled rip currents fluctuate";
(2) Abstract Line 26-27: remove statement "using Boussinesq model shows its superiority in study fine-scale..."
(3) Section 4.1 Line 239-241: Add statement "This feature is also visible in the CCTV images of the Dadonghai Beach (Fig. 9 in Li et al., 2016 and Fig. 6 in Wang et al., 2018) in the same location".
(4) Section 4.1 Line 262-265: Add statement "Although using Boussinesq model shows its superiority in studying fine-scale nearshore circulation and its variability in surf zones (Shi et al., 2012; Chen etal., 1999, 2003; Geiman et al., 2011; Fedderson, 2014; Zhang et al., 2021), it should

be noted that further analysis on rip pulsation and variability in this study is inappropriate due to lack of field observations at the Dadonghai Beach."

(5) Conclusion Line 465-466: It should be noted that lack of field observations largely limits our analysis of rip current dynamics at the Dadonghai Beach. Besides, sensitivity study on assignment of wave-following coefficient of virtual swimmers cf showcases large uncertainties of the Lagrangian tracking simulations in the study, which also urges a comprehensive field campaign on rip currents and associated hazards in the future at this rip-prone area.

**4. Where is the wave point located? It is essential to know its location in order to describe the wave conditions. This point should be indicated on the location map in Figure 1. Why not evaluate the entire wave time series considering the 30 years characterizing summers and winters and then specifically 2017-2018?**

**Response:** All accepted. The wave point is located at 18.2N, 109.5E, which is marked as a yellow triangle in Figure 1a (Page 21 in revised manuscript with tracked change). Wave rose diagram of 30-year-long time series of wave hindcast off the Dadonghai Bay is included in Figure 2a (Page 22), which is also shown in Figure R5 in this response. *Section 2.2 Wave conditions* (Line 142-162) has been reformulated as follows:

"Based on an analysis of 30-year wave hindcast dataset developed by National Marine Environmental Forecasting Center of China (NMEFC), wave conditions immediately off the Dadonghai Bay are assessed. The wave point for analysis is located at 109.5 E, 18.2 N, which is denoted as a yellow triangle in Fig. 1a. According to wave rose diagram of 30-year-long wave hindcast (Fig. 2a), generally the Dadonghai Bay receives waves at 2 prevailing directions, including powerful typhoon swells mainly from the south, and monsoon swells from the southeast. The 1-day moving average on the hourly wave hindcast in 2018 is performed and shown in Fig. 2d for further analysis. Waves during the summer months are relatively more energetic than in winter immediately off the Dadonghai Bay, which is interspersed with high-energy events associated with typhoon activities in the northern South China Sea (SCS). Typhoons can send long-period swells to the Dadonghai Bay. Wave-buoy observations along the slope of the Northern SCS demonstrate that while peak wave period during the winter monsoon is between 4 - 8 second, it is capable to reach up to 10 - 14 second during the passage of the tropical cyclones in summer (Xu et al., 2017; Tian et al., 2020). As shown in Fig. 2c, these storm waves arrive over a wide spread of directions from SE to SW. As typhoons enter the northern SCS and move west, the directions of waves receive by the Dadonghai Bay vary over time. Two prominent peaks in July and August are explained by two typhoons moving westward through the Northern SCS in 2018 summer, leading to elevated significant wave height of 1.5 - 2.5 and 2.0 - 4.0 m (without 1-day moving average), respectively. Strong typhoon swells persist for a week. During the winter months, the prevailing winter monsoon sometimes produce northeast wind waves in the Northern SCS with significant wave height of more than 3 m. However, as the monsoon swells propagate into the bay from the open sea, the waves get weaker in wave height and are diffracted to the southeast due to shelter of the peninsula to the east (Fig. 2b). The hourly significant wave height off the Dadonghai Bay generally varies below 1.0 m, and can reach up to 2.0 m with the outbreak of monsoon.

Accordingly, various wave conditions representative of winter monsoon swells and summer typhoon swells are used in the following modeling studies of rip variability and hazards."

[Figure]

Figure R5 Wave climate in study site. Subplot (a) is wave rose diagram of 30-year-long time series of wave hindcast at a wave point immediately off the Dadonghai Bay (109.5 E, 18.2 N); (b-c) are wave rose diagrams for winter monsoon and summer typhoon seasons in 2018, respectively; (d) is 1 day-averaged time series of significant wave height in 2018.

*Details:*

**1. The abstract is quite long and it is preferable to be more concise in order to attract the readers' attention.**

**Response:** Accepted. The abstract has been shortened. Please refer to the revised manuscript.

**2. Paragraph between Line 108 and 112 is not a description of the bathymetry. It should be included as another subsection describing sea levels.**

**Response:** Accepted. Indeed it is inappropriate to include this paragraph in Section 2.1. Instead of starting another subsection about tidal levels, we removed this paragraph from Section 2.1 and merged it into the opening paragraph of Section 2. Please refer to Line 99-101 in the revised manuscript.

**3. Table 1: How were these 9 tests defined? Are they the most frequent characteristics of summers? If the selected tests are representative of summer wave conditions, it would be appropriate to deepen this analysis in section 2.2 by considering the complete wave time series.**

**Response:** Accepted. *Section 2.2 Wave conditions* has been reformulated. Wave conditions given in 9 tests are representative of winter monsoon swells and summer typhoon swells. Among these wave conditions, $Hs$ = 4.5 and 8.0 s are representative of short-period monsoon swells in winter, and $Hs$ = 12.0 s is representative of long-period typhoon-induced swells in summer. Wave incident angles of 5° and 20° is also adopted to represent oblique SE or SW swells. Accordingly, the 2nd paragraph of *Section 3.2 Hydrodynamic settings* has been slightly revised (Line 195-197).

> "Tidal elevation is considered by adjusting input bathymetry according to averaged tidal range at the Dadonghai Beach. Wave conditions given in these 9 tests are representative of summer typhoon swells and winter monsoon swells, and hereafter referred to as T1 - T9."

**6. Line 227 - 235: Do these diagrams consider the results of all simulations in all tests?**

**Response:** Statistical analysis of Test 1, 7 and 8 is not included in Figure 5. The generated rip currents are weak in Test 1 with wave forcing of $Hs$=0.7 m and $Tp$=12 s. Test 7 and 8 are aimed to test the effect of wave incident angle on rip currents. The modeling results show that nearshore circulation is dominated by persistent longshore currents. We made a short statement in the caption of Figure 5 to explain that T1 and T7-8 were not included (Page 25 in the revise manuscript ).

*Editorial errors:*

*Line 42: exist => exit;*
*Line50: determine => determines;*
*Line 53: hazard => hazards;*
*Line 54: How => how, level => level of;*
*Line 94: beach => Beach;*
*Line 110: coastal coral reef => a coastal coral reef;*
*Line 112: been => be;*
*Line 121: add 'the' before NMEFC;*
*Line 124: averaging => average;*
*Line 140: In our simulation => In this study;*

*Table 1: defining  λ;*

*Line 155: are => is;*
*Line 157: add 'the' before 'hydrodynamic response';*
*Line 166: We define rip currents => In this study the rip currents are defined as...;*
*Line 171: add 'the' before 'greatest danger';*
*Line 172: location => locations;*
*Line 176: add 'the' before 'moverment';*
*Line 184: add 'a' before 'safe state';*
*Line 195: move => moves;*
*Line 204: deflect => deflects*
*Line 246: an => a;*
*Line 285: remove space;*
*Line 298: child => a child;*
*Line 312: to Child => for children;*
*Line 342 & 353: AREA => Area;*

*Line 401: Floating => floating;*

**Response:** All the editorial errors have been corrected in the revised manuscript.

*Figure improvement:*

*Figure1: Indicate geographical north orientation; Define zoom frame panels b and d; Include the location of the wave series point.*

**Response:** partly accepted. In Figure 1 (Page 21), symbols for geographical north orientation and wave point have been added. However, scale in Figure 1a is too large to fit the zoom frame for panels b and d.

*Figure 2: Preferable to indicate time in actual dates.*

**Response:** Accepted. In Figure 2 (Page 22), the julian day is replaced with month names.

*Figure 3 & 4: (1)The "s" in Hs does not need to be subscripted; (2)It is recommended to superimpose guide vectors over the currents to highlight the predominance; (3)Why is the wave direction specified only in the last 2 panels and not in the rest? (4)Panels e to i would correspond to tests 1 to 8. Where would the result of test 9 be?*

**Response:** In figure 3&4 (Page 23-24): (1) The bulk parameter Peak Significant Wave Height is denoted with $H_s$ in the study. We also checked with the related literature. Most of them subscript letter 's' to denote significant wave height. (2) There were several versions of Figure 3&4 before we submitted the manuscript. Subplots with guided vectors was one of them. However, each row of subplots in the figure use different length scales of vectors, and labeling them individually may cause confusion. Finally, we decided to use shaded contours overlapped with vectors to highlight the surf-zone flows. Here the vector plots are just for illustration.(3) All the parameters defining wave conditions are listed in each subplot. (4) Test 9 shows how variation in tidal level affect rip currents. The result is illustrated in Figure 5f. As the nearshore circulation is too weak in Test 9, we haven't included it in Figure 3&4.

*Figure 7: Specify which is this test indicating wave direction and tide level.*

**Response:** Accepted. In Figure 7 (Page 27), figure caption has been revised as "Rip hazard maps for shore-normal incident waves of $Hs$ = 1.0 m, and $Tp$ = 12 s (corresponding to T2 with mean tidal level) at the Dadonghai Beach, using bathymetry on August 7, 2018 (a) and December 26, 2019 (b)."

**Reference:**

(1) Castelle B., G. Masselink. Morphodynamics of wave-dominated beaches. Cambridge Prisms: Coastal Futures, 2023, 1, pp.e1. ff10.1017/cft.2022.2ff. ffhal-03830565

(2) Daly,C. J., C. Winter, Karin R. Bryan, On the morphological development of embayed beaches, Geomorphology, 248, 252-263, 2015.

(3) Chen, Q., Dalrymple, R. A., Kirby, J. T., Kennedy, A. and Haller, M. C., 1999, ``Boussinesq modeling of a rip current system'' , Journal of Geophysical Research, 104, 20,617 - 20, 637.

(4) Chen, Q., Kirby, J. T., Dalrymple, R. A., Shi, F., & Thornton, E. B. (2003). Boussinesq modeling of longshore currents. Journal of Geophysical Research, 108(11), 3362.

(5) Feddersen, F., 2014: The Generation of Surfzone Eddies in a Strong Alongshore Current. J. Phys. Oceanogr., 44, 600–617, https://doi.org/10.1175/JPO-D-13-051.1.

(6) Geiman, J. D., and J. T. Kirby, 2013: Unforced Oscillation of Rip-Current Vortex Cells. J. Phys. Oceanogr., 43, 477–497, https://doi.org/10.1175/JPO-D-11-0164.1.

(7) Hsu, J. R-C.; Evans, C. Parabolic bay shapes and applications. In: Proceedings of Institution of Civil Engineers - Part 2. London: Thomas Telford, 1989.

(8)Li, Z.: Rip current hazards in South China headland beaches, Ocean and Coastal Management, 121, 23–32, https://doi.org/10.1016/j.ocecoaman.2015.12.005, 2016.

(9) Silvester, R., John R. C. Hsu. Coastal Stabilization. World Scientific, 1997

(10) Tian, D.; Zhang, H.; Zhang, W.; Zhou, F.; Sun, X.; Zhou, Y.; Ke, D. Wave Glider Observations of Surface Waves During Three Tropical Cyclones in the South China Sea. Water 2020, 12, 1331. https://doi.org/10.3390/w12051331

(11) Xu, Y., H. He, J. Song, Y. Hou, and F. Li, 2017: Observations and Modeling of Typhoon Waves in the South China Sea. J. Phys. Oceanogr., 47, 1307–1324, https://doi.org/10.1175/JPO-D-16-0174.1.

(12) Wang, H. et al., 2018. Numerical simulations of rip currents off arc-shaped coastlines, Acta Oceanologica Sinica, 37(3): 21-30, https://doi.org/10.1007/s13131-018-1197-1},

(13) Wang, Ying & Martini, I. Peter & Zhu, Dakui & Zhang, Yongzhan & Tang, Wenwu. (2001). Coastal plain evolution in southern Hainan Island, China. Chinese Science Bulletin. 46. 90-96. 10.1007/BF03187244.

**Response to referee #2:**

*1. The paper presents a Boussinesq model study of rip current variability and the evaluation of hazards induced by rip currents. The study showcases the trajectory tracking method used in an embayed beach area for swimmer escape strategies. It is an interesting study.*

*Because the first author was the developer of FUNWAVE-GPU, I don't have many concerns about modeling details. However, I feel it may need logical consistency across the entire article, from the introduction to the conclusions. The authors expressed the importance of a wave-resolving model for such a trajectory-tracking study due to the random and dynamic nature of rip currents. But the paper concluded that the results from the Boussinesq model are comparable to that from the wave-averaged method. If the swimmer escape strategies are the main objective of the study, using the expensive Boussinesq model seems to be an overkill. For this reason, I suggest that the authors may emphasize more about the effects of IG and VLF bands because a regular wave averaged model cannot predict IG motions without using a non-stationary wave condition. The VLF motions in the model results are interesting. It is good to check if those are the shear wave mode as said or long gravity wave oscillations. A plot of a wavenumber spectrum would be helpful.*

*In general, the paper was well-written and easy to follow. It should be published after addressing the issue mentioned above.*

**Response:** Thanks for the insightful comments.
First of all, we would like to clarify that the 'wave-averaged velocity' in the manuscript (i.e., Abstract, Section 5.2) is computed by averaging phase-resolving velocities in two wave periods (24 seconds) in FUNWAVE-TVD, rather than the velocity yield by other short-wave-averaged models such as XBeach-SB (XBeach-SurfBeat). Both the wave-resolving or wave-averaged tracking in the study all retained the non-stationary IG or VLF motions, thus presenting similar trajectories of virtual swimmers. We would like to reiterate here that the wave-resolving tracking means using instanteneous, random velocities at each timestep (~0.04 s) to tracer swimmers, while wave-averaged tracking uses 24 s-averaged mean flow velocity to tracer swimmers.

Previous beach-safety studies (McCarroll et al., 2015; Castelle et al., 2016) use XBeach-SB to study rip dynamics and Lagrangian tracking of virtual swimmers. XBeach has 2 different dynamic frameworks inside, namely 'Surf Beat (SB)' and 'Non-hydrostatic (NH)'. Though XBeach-SB is a phase-averaged surfzone model, it offers an additional advantage compared to other phase-averaged models in its ability to resolve wave-group generated infragravity (IG) motions, vortical currents at VLF timescales. XBeach-SB uses the 2D wave spectra at its offshore boundary to calculate the wave group envelope, and then solves the variation of short-waves envelope (wave height) on the scale of wave groups. This variation in turn drives IG waves within the depth-integrated hydrostatic solver, approximating transient behaviors and swash dynamics in the nearshore zone. The non-hydrostatic module (XBeach-NH) extends XBeach's capability to wave-resolving modeling of non-linear waves, wave-current interaction and wave breaking in the surf zone. Both Boussinesq-type wave model FUNWAVE and XBeach-NH are phase-resolving. The fundamental difference between them is that FUNWAVE is a depth-integrated model, and relies on higher-order derivative terms to improve the frequency dispersion, while it is improved in NH models by including vertical layers.

There was a comparison between power spectra of incoming waves and resulting flow fluctuations at a given point in Figure 6b. We have also set a list of output points along the beach, whose power spectra were not shown in the manuscript. The spectra demonstrated that the wave-group-forced IG waves were persistent at all points, though their amplitudes varied from point to point; while Very low-frequency [VLF, O(10 min)] motions only existed at several points where rip flows pulsated in amplitude and directions. The absence of spectral peak of incoming waves at VLF band suggested that the VLF motions were not forced by the long-period component of gravity waves. Therefore, we concluded that the VLF motions were usually related to the formation of vortex due to local morphology. The detailed analysis of generation mechanism of VLF motions was not available in the manuscript due to lack of in-situ instrumental observations.

To avoid misunderstanding, we have made several revisions in the manuscript:

(1) Abstract Line25-27: the statement "Virtual trajectories yielded by the wave-resolving and wave-averaged velocities are generally consistent with each other" is removed;

(2) Introduction Line 89-90: remove "Discussion has been made to highlight how the phase-resolving and random trajectories of swimmers are different with those using time-averaged velocities."

(3) Introduction Line 94-95: remove "with further discussion on wave-resolving tracking of swimmers given in Section 5"

(4) Section 5.2 Line 396, Line 404: "wave-averaged velocity" is replaced with "mean flow velocity";

(5) Section 5.2 Line 407-411: The sentence is rewritten as "Fig. 13a-b suggest that resolving wave scale in the Lagrangian tracking produces comparable statistical results of escape time $t_{safe}$ with that using the mean-flow velocity in general. Nevertheless, effect of wave randomness on individual trajectories can be observed, which results in different floating paths for a few virtual swimmers (denoted by long orange line segments)."

(6) Section 5.2 Line 412-415: The whole paragraph is removed;

(7) Conclusion Line 464-465: Here we confine our conclusion on the effect of wave randomness on the trajectories of virtual tracers: "Wave-resolving Lagrangian tracking helps us understand the effect of wave randomness on surf-zone flows and bathers' trajectories."

*Details:*

**2. Figure 1(f), please provide the vertical datum for 2018 and 2019 measurements. Provide an accuracy estimate for the satellite-derived bathymetric data.**

**Response:** Accepted.

(1) Two satellite images were acquired from Google Earth Historical Database on August 7, 2018 and December 26, 2019 when sea state was calm. However, the exact collecting time of the images was not clear to us. Therefore, the tidal levels when the satellite images were acquired could not be precisely determined from the nearly tidal gauge. According to the tidal records of nearly Sanya tidal gauge, Dadonghai Beach has a micro-tidal range, and tidal ranges on August 7, 2018 and December 26, 2019 were approximately 0.97 m and 1.62 m. The collecting time of the images were roughly estimated by

the shadows of the tall buildings along the beach. We speculated that the picturing time on August 7, 2018 was around early morning when the projected shadows were elongated and on the west side of the building; while on December 26, 2019 it was at the mid-day. According to this information, the tidal levels were around mid-to-high tide and mid-tide, respectively. The computation grids were constructed by combining inversion of satellite images along the beach and nautical chart within the Dadonghai Bay. The combined bathymetry were converted to the mean tidal level (MTL) based on the datum at Sanya Tidal gauge nearby.

(2)In this study the detailed echosounder survey covering the Dadonghai Bay was not performed. Beach bathymetries were inverted from two satellite images acquired from Google Earth Historical Database on August 7, 2018 and December 26, 2019 when sea state was calm. The commonly-used techniques which allow indirect measurement of depth by satellite images are based on water color and wave kinematics. Colour-based methods are usually limited to non-turbid or non-breaking waters. The wave kinematics methods work well in turbid or optically deep water. While remote-sensed bathymetry is not as accurate as in-situ measurement, it is easily available by satellite and UAV, making it desirable for operational forecasting of rip hazards.

The depth data for the inversion were from two sources, including nautical charts and field survey. For main body of the Dadonghai Bay, large-scale nautical chart (1:25,000, purchased from China Navigation Press) was interpolated and converted to the mean tidal level (MTL) based on the datum at Sanya Tidal gauge nearby, which was then correlated against the corresponding color pixels.

Surf-zone bathymetry is constantly changing in multiple temporal scales. For surf zone of the Dadonghai Beach that was not covered by the nautical chart, we relied on some in-situ photos, UAV images and public surveys collected from a field trip of Dadonghai Beach conducted in October 2019 by National Marine Hazard Mitigation Service of State Oceanic Administration (NMHMS). These information and images were combined to derive appropriate color-depth relation in the surf zone. During the survey, an UAV was released to observe beach morphology. As shown in Figure R1c (courtesy of NMHMS), well-developed, periodic crescent sandbars were recorded by the UAV, whose locations coincided with those appeared in the satellite image (Figure R1b). According to photos and public surveys of swimmers and coastal patrols collected during the field trip, water depth varied

approximately between 0.5－1.0 m over the shore-connected sandbars, and 1.5－2.0 m over the rip

channels. Floating separation ropes were deployed perpendicularly to the beach by coastal patrols to prevent bathers from swimming to deeper rip channels.

In general, as suggested by field data, though no precise bathymetric measurement was conducted in the study, the major features of beach morphological could be reflected by the satellite images, which are thought to be adequate for models to grasp the main rip patterns.

In the revised manuscript, we re-structured 2nd paragraph of *Section 2.1 Surf-zone bathymetry* as follows.

> "Compared to extensive sonar or in-situ measurements of depth, shallow bathymetry can be fast and cost-effectively evaluated by remote-sensing images. In this study, by establishing a site-specific linear relationship between pixel colors and depths, nearshore bathymetry at Dadonghai was

mapped and interpolated to 1-meter resolution. Although this inversion may not produce bathymetry as accurate as other approaches, it can be operationalized for rip hazard forecast in future owing to its simplicity to locate sandbars and shoals, as well as availability of satellite imagery (Radermacher et al., 2018). It should be noted that the in-situ echosounder survey was not performed in August 2018 and December 2019 for this study. For main body of the Dadonghai Bay, the depth data for the color-depth correlation was from nautical chart published by China Navigation Press with a scale of 1:25000, which was converted to the mean tidal level (MTL) based on the datum at Sanya Tidal gauge nearby. Surf-zone bathymetry is constantly changing in multiple temporal scales. For the surf zone that is not covered by the nautical chart, only very limited field data collected in October 2019 was available to derive the color-depth relation along the surf zone. The derived bathymetry was then rotated 90 degrees to align the shoreline with the vertical axis. The rotation is necessary for FUNWAVE to apply irregular wave maker.

[Figure]

Figure R1. Crescent sandbars recorded in the satellite (b) and UAV (c) images, respectively. The satellite image was collected on December 13, 2019 (Google Earth Historic Imagery), and the UAV image was collected during the rip-hazard field investigation at the same period on December 9, 2019 (Courtesy of National Marine Hazard Mitigation Service, China).

***3. How does a periodic boundary condition set up in such a bay-like domain?***

**Response:** Accepted. In the study a south open boundary is placed in the grid, and there is no east-west open boundaries. Therefore, lateral periodic boundary condition is not necessary. The expression in Line 128-129 is revised accordingly as follows.

"The rotation is necessary for FUNWAVE to apply irregular wave maker and periodic boundary" => "The rotation is necessary for FUNWAVE to apply internal wavemaker".

***4. Line 112, both ends is exposed, grammar***

**Response:** Accepted. Line 117, this sentence has been removed.

***5. Line 139, FUNWAVE-GPU add the reference here***

**Response:** Accepted. The reference *Yuan et al., 2020* has been added in Line 169.

***6. Line 142, CFL, add the complete terminology***

**Response:** Accepted. In Line 173, 'CFL' is replaced with 'Courant–Friedrichs–Lewy (CFL)' when first appeared.

*7. Line 144, friction coefficient, need to clarify bottom friction form, manning formula? or provide a reference*

**Response**: Accepted. In Line 175, "The bottom friction coefficient is 0.025" is replaced with "A constant bottom drag coefficient of 0.0025 in the quadratic friction formula was applied". We added a reference of Zhang et al., 2022 in the manuscript.

> Zhang, Y., Shi, F., Kirby, J. T., & Feng, X. (2022). Phase-resolved modeling of wave interference and its effects on nearshore circulation in a large ebb shoal-beach system. Journal of Geophysical Research: Oceans, 127, e2022JC018623. https://doi.org/10.1029/2022JC018623

*8. Section 3.3. It's interesting to make a definition for the hazard levels. Any reference for this definition, or just created by the authors?*

**Response:** The rip hazard levels are created by the authors. It is preferable to quantify and visualize the rip hazard by simple and effective indexes or guidelines for operational centers or coastal patrol. Thus, we proposed an index table for rip hazard levels by combining rip strength and duration.

*9. Line 181, arbitrary factor of 0.8. need a sensitivity test on this number*

**Response:** Accepted. We have made a sensitivity study in the arbitrary factor (0.8) in Equation 1 ($u_{tracking} = c_f u_w + U_s$, where $c_f$ is defined as wave-following factor; $u_{tracking}$ is Lagrangian tracking velocity, $u_w$ is instantaneous wave-induced rip flow, and $U_s$ is swimming velocity). Here $c_f$ defines how well a tracer (virtual swimmer) drifts with the ambient flow.

We have opened a new subsection *Section 5.3* in the revised manuscript, and results of the sensitivity study by varying the factor from 0.4 to 1.0 are included. An brief analysis is included here.

This arbitrary factor is defined as the wave-following factor varying from 0.4 to 1.0 with an interval of 0.2. As shown in Figure R2, We mainly focus on the swimmers that do not reach safety after 10-min swimming onshore (trajectories with red color). In the case of $c_f = 0.4$, almost entire seeded swimmers can get safe by swimming onshore with an average swimming velocity of 0.2 m/s, even for swimmers that are seeded in outer surf zone. However, at the opposite extreme, more than 40% of swimmers are exhausted in the rip eddy and fail to reach safe areas when the wave-following factor of 1.0 is set in the model.

The sensitivity study suggests that the value of wave-following factor is crucial to the tracking results of virtual swimmers, which in turn influences swimmer escape strategies. The factor should be calibrated in the further field studies.

**Swim onshore**

[Figure]

Figure R2. Sensitivity study of floating factor in Equation 1. Histograms of $t_{safe}$ give percentages of swimmers who have reached safety at each $t_{safe}$ range. The factor varies from 0.4 to 1.0 with an interval of 0.2 to define how well swimmers float with the ambient wave-induced flows (a-d). Assigning 0.4 means that swimmers are only slightly affected by surf-zone flows, and assigning 1.0 means that swimmers float with the ambient water perfectly. In this case, virtual swimmers have a constant onshore swimming velocity of 0.2 m/s.

**Reference:**

(1) Castelle, B., McCarroll, R., Brander, R., T., S., and B., D.: Modelling the alongshore variability of optimum rip current escape strategies on a multiple rip-channelled beach, Nat Hazards, 81, 663–686, https://doi.org/10.1007/s11069-015-2101-3, 2016.

(2) McCarroll, R. J., Castelle, B., Brander, R., and T., S.: Modelling rip current flow and bather escape strategies across a transverse bar and rip channel morphology, Geomorphology, 246, 502–518, https://doi.org/10.1016/j.geomorph.2015.06.041, 2015.

(3) Salatin, Reza & Chen, Qin & Bak, A. & Shi, Fengyan & Brandt, Steven. (2021). Effects of Wave Coherence on Longshore Variability of Nearshore Wave Processes. Journal of Geophysical Research: Oceans. 126. 10.1029/2021JC017641.

(4) Zhang, yu & Shi, Fengyan & Kirby, James & Feng, Xi. (2022). Phase-Resolved Modeling of Wave Interference and Its Effects on Nearshore Circulation in a Large Ebb Shoal-Beach System. Journal of Geophysical Research: Oceans. 127. 10.1029/2022JC018623.

---

## Author Response (AR2)

**Authors' response to the Editor and Referees**
**on**
**egusphere-2023-122 "A wave-resolving modeling study of rip current variability, rip hazard, and swimmer escape strategies on an embayed beach with irregular rip channels"**

Dear Prof. Mauricio Gonzalez and Referees,

We are really delighted that this study was accepted for publication, and we really appreciate the constructive comments raised by both reviewers. Thanks very much.

Kind Regards,
Huaiwei Yang and Ye Yuan, on behalf of the co-authors.